# Three Operator Splitting with Subgradients, Stochastic Gradients, and Adaptive Learning Rates

**Alp Yurtsever**[*]
Umeå University
alp.yurtsever@umu.se

**Alex Gu**[*]    &    **Suvrit Sra**
Massachusetts Institute of Technology
{gua,suvrit}@mit.edu

## Abstract

Three Operator Splitting (TOS) (Davis & Yin, 2017) can minimize the sum of multiple convex functions effectively when an efficient gradient oracle or proximal operator is available for each term. This requirement often fails in machine learning applications: *(i)* instead of full gradients only stochastic gradients may be available; and *(ii)* instead of proximal operators, using subgradients to handle complex penalty functions may be more efficient and realistic. Motivated by these concerns, we analyze three potentially valuable extensions of TOS. The first two permit using subgradients and stochastic gradients, and are shown to ensure a $\mathcal{O}(1/\sqrt{t})$ convergence rate. The third extension ADAPTOS endows TOS with adaptive stepsizes. For the important setting of optimizing a convex loss over the intersection of convex sets ADAPTOS attains universal convergence rates, *i.e.*, the rate adapts to the *unknown* smoothness degree of the objective function. We compare our proposed methods with competing methods on various applications.

## 1   Introduction

We study convex optimization problems of the form

$$\min_{x \in \mathbb{R}^n} \quad \phi(x) := f(x) + g(x) + h(x), \tag{1}$$

where $f : \mathbb{R}^n \to \mathbb{R}$ and $g, h : \mathbb{R}^n \to \mathbb{R} \cup \{+\infty\}$ are proper, lower semicontinuous and convex functions. Importantly, this template captures constrained problems via indicator functions. To avoid pathological examples, we assume that the relative interiors of $\mathrm{dom}(f)$, $\mathrm{dom}(g)$ and $\mathrm{dom}(h)$ have a nonempty intersection.

Problem (1) is motivated by a number of applications in machine learning, statistics, and signal processing, where the three functions comprising the objective $\phi$ model data fitting, structural priors, or decision constraints. Examples include overlapping group lasso (Yuan et al., 2011), isotonic regression (Tibshirani et al., 2011), dispersive sparsity (El Halabi & Cevher, 2015), graph transduction (Shivanna et al., 2015), learning with correlation matrices (Higham & Strabić, 2016), and multidimensional total variation denoising (Barbero & Sra, 2018).

An important technique for addressing composite problems is *operator splitting* (Bauschke et al., 2011). However, the basic proximal-(sub)gradient method may be unsuitable for Problem (1) since it

---

[*]Alp Yurtsever and Alex Gu contributed equally to this paper. The paper is based primarily on the work done while Alp Yurtsever was at Massachusetts Institute of Technology.

requires the prox-operator of $g + h$, computing which may be vastly more expensive than individual prox-operators of $g$ and $h$. An elegant, recent method, Three Operator Splitting (TOS, Davis & Yin (2017), see Algorithm 1) offers a practical choice for solving Problem (1) when $f$ is smooth. Importantly, at each iteration, TOS evaluates the gradient of $f$ and the proximal operators of $g$ and $h$ only once. Moreover, composite problems with more than three functions can be reformulated as an instance of Problem (1) in a product-space and solved by using TOS. This is an effective method as long as each function has an efficient gradient oracle or proximal operator (see Section 2).

Unfortunately, TOS is not readily applicable to many optimization problems that arise in machine learning. Most important among those are problems where only access to stochastic gradients is feasible, *e.g.*, when performing large-scale empirical risk minimization and online learning. Moreover, prox-operators for some complex penalty functions are computationally expensive and it may be more efficient to instead use subgradients. For example, proximal operator for the maximum eigenvalue function that appears in dual-form semidefinite programs (*e.g.*, see Section 6.1 in (Ding et al., 2019)) may require computing a full eigendecomposition. In contrast, we can form a subgradient by computing only the top eigenvector via power method or Lanczos algorithm.

**Contributions**. With the above motivation, this paper contributes three key extensions of TOS. We tackle nonsmoothness in Section 3 and stochasticity in Section 4. These two extensions enable us to use subgradients and stochastic gradients of $f$ (see Section 2 for a comparison with related work), and satisfy a $\mathcal{O}(1/\sqrt{T})$ error bound in function value after $T$ iterations. The third main contribution is ADAPTOS in Section 5. This extension provides an adaptive step-size rule in the spirit of AdaGrad (Duchi et al., 2011; Levy, 2017) for an important subclass of Problem (1). Notably, for optimizing a convex loss over the intersection of two convex sets, ADAPTOS ensures universal convergence rates. That is, ADAPTOS implicitly adapts to the *unknown* smoothness degree of the problem, and ensures a $\tilde{\mathcal{O}}(1/\sqrt{t})$ convergence rate when the problem is nonsmooth but the rate improves to $\tilde{\mathcal{O}}(1/t)$ if the problem is smooth and a solution lies in the relative interior of the feasible set.

In Section 6, we discuss empirical performance of our methods by comparing them against present established methods on various benchmark problems from COPT Library (Pedregosa et al., 2020) including the overlapping group lasso, total variation deblurring, and sparse and low-rank matrix recovery. We also test our methods on nonconvex optimization by training a neural network model. We present more experiments on isotonic regression and portfolio optimization in the supplements.

**Notation.** We denote a solution of Problem (1) by $x_\star$ and $\phi_\star := \phi(x_\star)$. The distance between a point $x \in \mathbb{R}^n$ and a closed and convex set $\mathcal{G} \subseteq \mathbb{R}^n$ is $\mathrm{dist}(x, \mathcal{G}) := \min_{y \in \mathcal{G}} \|x - y\|$; the projection of $x$ onto $\mathcal{G}$ is given by $\mathrm{proj}_{\mathcal{G}}(x) := \arg\min_{y \in \mathcal{G}} \|x - y\|$. The prox-operator of a function $g : \mathbb{R}^n \to \mathbb{R} \cup \{+\infty\}$ is defined by $\mathrm{prox}_g(x) := \arg\min_{y \in \mathbb{R}^n} \{g(y) + \frac{1}{2}\|x - y\|^2\}$. The indicator function of $\mathcal{G}$ gives $0$ for all $x \in \mathcal{G}$ and $+\infty$ otherwise. Clearly, the prox-operator of an indicator function is the projection onto the corresponding set.

## 2 Background and related work

TOS, proposed recently by Davis & Yin (2017), can be seen as a generic extension of various operator splitting schemes, including the forward-backward splitting, Douglas-Rachford splitting, forward-Douglas-Rachford splitting (Briceño-Arias, 2015), and the generalized forward-backward splitting (Raguet et al., 2013). It covers these aforementioned approaches as special instances when the terms $f, g$ and $h$ in Problem (1) are chosen appropriately. Convergence of TOS is well studied when $f$ has Lipschitz continuous gradients. It ensures $\mathcal{O}(1/t)$ convergence rate in this setting, see (Davis & Yin, 2017) and (Pedregosa, 2016) for details.

Other related methods that can be used for Problem (1) when $f$ is smooth are the primal-dual hybrid gradient (PDHG) method (Condat, 2013; Vũ, 2013) and the primal-dual three operator splitting methods in (Yan, 2018) and (Salim et al., 2020). These methods can handle a more general template where $g$ or $h$ is composed with a linear map, however, they require $f$ to be smooth. The convergence rate of PDHG is studied in (Chambolle & Pock, 2016).

*Nonsmooth setting.* We are unaware of any prior result that permits using subgradients in TOS (or in other methods that can use the prox-operator of $g$ and $h$ separately for Problem (1)). The closest match is the proximal subgradient method which applies when $h$ is removed from Problem (1), and it is covered by our nonsmooth TOS as a special case.

*Stochastic setting.* There are multiple attempts to devise a stochastic TOS in the literature. Yurtsever et al. (2016) studied Problem (1) under the assumption that $f$ is smooth and strongly convex, and an unbiased gradient estimator with bounded variance is available. Their stochastic TOS has a guaranteed $\mathcal{O}(1/t)$ convergence rate. In (Cevher et al., 2018), they drop the strong convexity assumption, instead they assume that the variance is summable. They show asymptotic convergence with no guarantees on the rate. Later, Pedregosa et al. (2019) proposed a stochastic variance-reduced TOS and analyzed its non-asymptotic convergence guarantees. Their method gets $\mathcal{O}(1/t)$ convergence rate when $f$ is smooth. The rate becomes linear if $f$ is smooth and strongly convex and $g$ (or $h$) is also smooth. Recently, Yurtsever et al. (2021) studied TOS on problems where $f$ can be nonconvex and showed that the method finds a first-order stationary point with $\mathcal{O}(1/\sqrt[3]{t})$ convergence rate under a diminishing variance assumption. They increase the batch size over the iterations to satisfy this assumption.

None of these prior works cover the broad template we consider: $f$ is smooth or Lipschitz continuous and the stochastic first-order oracle has bounded variance. To our knowledge, our paper gives the first analysis for stochastic TOS without strong convexity assumption or variance reduction.

Other related methods are the stochastic PDHG in (Zhao & Cevher, 2018), the decoupling method in (Mishchenko & Richtárik, 2019), the stochastic primal-dual method in (Zhao et al., 2019), and the stochastic primal-dual three operator splitting in (Salim et al., 2020). The method in (Zhao et al., 2019) can be viewed as an extension of stochastic ADMM (Ouyang et al., 2013; Azadi & Sra, 2014) from the sum of two terms to three terms in the objective. Similar to the existing stochastic TOS variants, these methods either assume strong convexity or require variance-reduction.

*Adaptive step-sizes.* The standard writings of TOS and PDHG require the knowledge of the smoothness constant of $f$ for the step-size. Backtracking line-search strategies (for finding a suitable step-size when the smoothness constant is unknown) are proposed for PDHG in (Malitsky & Pock, 2018) and for TOS in (Pedregosa & Gidel, 2018). These line-search strategies are significantly different than our adaptive learning rate. Importantly, these methods work only when $f$ is smooth. They require extra function evaluations, and are thus not suitable for stochastic optimization. And their goal is to estimate the *smoothness constant*. In contrast, our goal is to design an algorithm that adapts to the unknown *smoothness degree*. Our method does not require function evaluations, and it can be used in smooth, nonsmooth, or stochastic settings.

At the heart of our method lie adaptive online learning algorithms (Duchi et al., 2011; Rakhlin & Sridharan, 2013) together with online to offline conversion techniques (Levy, 2017; Cutkosky, 2019). Similar methods appear in the literature for other problem templates with no constraint or a single constraint in (Levy, 2017; Levy et al., 2018; Kavis et al., 2019; Cutkosky, 2019; Bach & Levy, 2019). Our method extends these results to optimization over the intersection of convex sets. When $f$ is nonsmooth, ADAPTOS ensures a $\tilde{\mathcal{O}}(1/\sqrt{t})$ rate, whereas the rate improves to $\tilde{\mathcal{O}}(1/t)$ if $f$ is smooth and there is a solution in the relative interior of the feasible set.

**TOS for more than three functions.** TOS can be used for solving problems with more than three convex functions by a product-space reformulation technique (Briceño-Arias, 2015). Consider

$$\min_{x \in \mathbb{R}^d} \quad \sum_{i=1}^{q} \phi_i(x), \tag{2}$$

where each component $\phi_i : \mathbb{R}^d \to \mathbb{R} \cup \{+\infty\}$ is a proper, lower semicontinuous and convex function. Without loss of generality, suppose $\phi_1, \ldots, \phi_p$ are prox-friendly. Then, we can reformulate (2) in the product-space $\mathbb{R}^{d \times (p+1)}$ as

$$\min_{(x_0, x_1, \ldots, x_p) \in \mathbb{R}^{d \times (p+1)}} \quad \sum_{i=1}^{p} \phi_i(x_i) + \sum_{i=p+1}^{q} \phi_i(x_0) \quad \text{subject to} \quad x_0 = x_1 = \ldots = x_p. \tag{3}$$

This is an instance of Problem (1) with $n = d \times (p+1)$ and $x = (x_0, x_1, \ldots, x_p)$. We can choose $g(x)$ as the indicator of the equality constraint, $f(x) = \sum_{i=p+1}^{q} \phi_i(x_0)$, and $h(x) = \sum_{i=1}^{p} \phi_i(x_i)$. Then, the (sub)gradient of $f$ is the sum of (sub)gradients of $\phi_{p+1}, \ldots, \phi_q$; $\text{prox}_g$ is a mapping that averages $x_0, x_1, \ldots, x_p$; and $\text{prox}_h$ is the concatenation of the individual prox-operators of $\phi_1, \ldots, \phi_p$.

To our knowledge, TOS has been studied only for problems with smooth $f$, and this forces us to assign all nonsmooth components $\phi_i$ in (2) to the proximal term $h$ in (3). In this work, by enabling subgradient steps for nonsmooth $f$, we provide the flexibility to choose how to process each nonsmooth component $\phi_i$ in (3), either by its proximal operator through $h$ or by its subgradient via $f$.

**Algorithm 1** Three Operator Splitting (TOS)

---

**Input:** Initial point $y_0 \in \mathbb{R}^n$, step-size sequence $\{\gamma_t\}_{t=0}^T$
**for** $t = 0, 1, 2, \ldots, T$ **do**
    $z_t = \text{prox}_{\gamma_t g}(y_t)$
    Choose an update direction $u_t \in \mathbb{R}^n$         $\{u_t = \nabla f(z_t)$ captures the standard version of TOS$\}$
    $x_t = \text{prox}_{\gamma_t h}(2z_t - y_t - \gamma_t u_t)$
    $y_{t+1} = y_t - z_t + x_t$
**end for**
**Return:** Ergodic sequence $\bar{x}_t$ and $\bar{z}_t$ defined in (5)

---

## 3 TOS for Nonsmooth Setting

Algorithm 1 presents the generalized TOS for Problem (1). It recovers the standard version in (Davis & Yin, 2017) if we choose $u_t = \nabla f(z_t)$ when $f$ is smooth. For convenience, we define the mapping

$$\text{TOS}_\gamma(y, u) := y - \text{prox}_{\gamma g}(y) + \text{prox}_{\gamma h}\big(2 \cdot \text{prox}_{\gamma g}(y) - y - \gamma u\big) \tag{4}$$

which represents one iteration of Algorithm 1.

The first step of the analysis is the fixed-point characterization of TOS. The following lemma is a straightforward extension of Lemma 2.2 in (Davis & Yin, 2017) to permit subgradients. The proof is similar to (Davis & Yin, 2017), we present it in the supplementary material for completeness.

**Lemma 1** (Fixed points of TOS). *Let $\gamma > 0$. Then, there exists a subgradient $u \in \partial f(\text{prox}_{\gamma g}(y))$ that satisfies $\text{TOS}_\gamma(y, u) = y$ if and only if $\text{prox}_{\gamma g}(y)$ is a solution of Problem (1).*

When $f$ is $L_f$-smooth, TOS with $u_t = \nabla f(z_t)$ is known to be an averaged operator[1] if $\gamma \in (0, 2/L_f)$ (see Proposition 2.1 in (Davis & Yin, 2017)) and the analysis in prior work is based on this property. In particular, averagedness implies Fejér monotonicity, *i.e.*, that $\|y_t - y_\star\|$ is non-increasing, where $y_\star$ denotes a fixed point of TOS. However, when $f$ is nonsmooth and $u_t$ is replaced with a subgradient, TOS operator is no longer averaged and the standard analysis fails. One of our key observations is that $\|y_t - y_\star\|$ remains bounded even-though we loose averagedness and Fejér monotonicity in this setting, see Theorem S.6 in the supplements.

**Ergodic sequence.** Convergence of operator splitting methods are often given in terms of ergodic (averaged) sequences. This strategy requires maintaining the running averages of $z_t$ and $x_t$:

$$\bar{x}_t = \frac{1}{t+1} \sum_{\tau=0}^t x_\tau \qquad \text{and} \qquad \bar{z}_t = \frac{1}{t+1} \sum_{\tau=0}^t z_\tau. \tag{5}$$

Clearly, we do not need to store the history of $x_t$ and $z_t$ to maintain these sequences. In practice, the last iterate often converges faster than the ergodic sequence. We can evaluate the objective function at both points and return the one with the smaller value.

We are ready to present convergence guarantees of TOS for the nonsmooth setting.

**Theorem 1.** *Consider Problem (1) and employ TOS (Algorithm 1) with the update directions and step-size chosen as*

$$u_t \in \partial f(z_t) \quad \text{and} \quad \gamma_t = \frac{\gamma_0}{\sqrt{T+1}} \text{ for some } \gamma_0 > 0, \quad \text{for } t = 0, 1, \ldots, T. \tag{6}$$

*Assume that $\|u_t\| \le G_f$ for all $t$. Then, the following guarantees hold:*

$$f(\bar{z}_T) + g(\bar{z}_T) + h(\bar{x}_T) - \phi_\star \le \frac{1}{2\sqrt{T+1}}\left(\frac{D^2}{\gamma_0} + \gamma_0 G_f^2\right) \tag{7}$$

$$\text{and} \quad \|\bar{x}_T - \bar{z}_T\| \le \frac{2}{T+1}\left(D + \gamma_0 G_f\right), \quad \text{where} \quad D = \max\{\|y_0 - x_\star\|, \|y_0 - y_\star\|\}. \tag{8}$$

---

[1]An operator $\text{T} : \mathbb{R}^n \to \mathbb{R}^n$ is $\omega$-averaged if $\|\text{T}x - \text{T}y\|^2 \le \|x - y\|^2 - \frac{1-\omega}{\omega}\|(x - \text{T}x) - (y - \text{T}y)\|^2$ for some $\omega \in (0, 1)$ for all $x, y \in \mathbb{R}^n$.

**Remark 1.** *The boundedness of subgradients is a standard assumption in nonsmooth optimization. It is equivalent to assuming that $f$ is $G_f$-Lipschitz continuous on $\mathrm{dom}(g)$.*

If $D$ and $G_f$ are known, we can optimize the constants in (7) by choosing $\gamma_0 = D/G_f$. This gives $f(\bar{z}_T) + g(\bar{z}_T) + h(\bar{x}_T) - \phi_\star \leq \mathcal{O}(DG_f/\sqrt{T})$ and $\|\bar{x}_T - \bar{z}_T\| \leq \mathcal{O}(D/T)$.

*Proof sketch.* We start by writing the optimality conditions for the proximal steps for $z_t$ and $x_t$. Through algebraic modifications and by using convexity of $f$, $g$ and $h$, we obtain

$$f(z_t) + g(z_t) + h(x_t) - \phi_\star \leq \frac{1}{2\gamma}\|y_t - x_\star\|^2 - \frac{1}{2\gamma}\|y_{t+1} - x_\star\|^2 + \frac{\gamma}{2}\|u_t\|^2. \tag{9}$$

$\|u_t\| \leq G_f$ by assumption. Then, we average this inequality over $t = 0, 1, \ldots, T$ and use Jensen's inequality to get (7).

The bound in (8) is an immediate consequence of the boundedness of $\|y_{T+1} - y_\star\|$ that we show in Theorem S.6 in the supplementary material:

$$\|y_{T+1} - y_\star\| \leq \|y_0 - y_\star\| + 2\gamma_0 G_f. \tag{10}$$

By definition, $\|\bar{x}_T - \bar{z}_T\| = \frac{1}{T}\|y_{T+1} - y_0\| \leq \frac{1}{T}(\|y_{T+1} - y_\star\| + \|y_\star - y_0\|)$. $\qquad\square$

Theorem 1 does not immediately yield convergence to a solution of Problem (1) because $f + g$ and $h$ are evaluated at different points in (7). Next corollary solves this issue.

**Corollary 1.** *We are interested in two particular cases of Theorem 1:*

(i). *Suppose $h$ is $G_h$-Lipschitz continuous. Then,*

$$\phi(\bar{z}_T) - \phi_\star \leq \frac{1}{2\sqrt{T+1}}\left(\frac{D^2}{\gamma_0} + \gamma_0 G_f^2\right) + \frac{2G_h}{T+1}\left(D + \gamma_0 G_f\right). \tag{11}$$

(ii). *Suppose $h$ is the indicator function of a convex set $\mathcal{H} \subseteq \mathbb{R}^n$. Then,*

$$f(\bar{z}_T) + g(\bar{z}_T) - \phi_\star \leq \frac{1}{2\sqrt{T+1}}\left(\frac{D^2}{\gamma_0} + \gamma_0 G_f^2\right) \tag{12}$$

$$and \quad \mathrm{dist}(\bar{z}_T, \mathcal{H}) \leq \frac{2}{T+1}\left(D + \gamma_0 G_f\right). \tag{13}$$

*Proof.* (i). Since $h$ is $G_h$-Lipschitz, $\phi(\bar{z}_T) \leq f(\bar{z}_T) + g(\bar{z}_T) + h(\bar{x}_T) + G_h\|\bar{x}_T - \bar{z}_T\|$.
(ii). $h(\bar{x}_T) = 0$ since $\bar{x}_T \in \mathcal{H}$. Moreover, $\mathrm{dist}(\bar{z}_T, \mathcal{H}) := \inf_{x \in \mathcal{H}}\|\bar{z}_T - x\| \leq \|\bar{z}_T - \bar{x}_T\|$. $\qquad\square$

**Remark 2.** *We fix time horizon $T$ for the ease of analysis and presentation. In practice, we use $\gamma_t = \gamma_0/\sqrt{t+1}$.*

Theorem 1 covers the case in which $g$ is the indicator of a convex set $\mathcal{G} \subseteq \mathbb{R}^n$. By definition, $\bar{z}_T \in \mathcal{G}$ and $x_\star \in \mathcal{G}$, hence $g(\bar{z}_T) = g(x_\star) = 0$. If both $g$ and $h$ are indicator functions, TOS gives an approximately feasible solution, in $\mathcal{G}$, and close to $\mathcal{H}$. We can also consider a stronger notion of approximate feasibility, measured by $\mathrm{dist}(\bar{z}_T, \mathcal{G} \cap \mathcal{H})$. However, this requires additional regularity assumptions on $\mathcal{G}$ and $\mathcal{H}$ to avoid pathological examples, see Lemma 1 in (Hoffmann, 1992) and Definition 2 in (Kundu et al., 2018).

Problem (1) captures unconstrained minimization problems when $g = h = 0$. Therefore, the convergence rate in Theorem 1 is optimal in the sense that it matches the information theoretical lower bounds for first-order black-box methods, see Section 3.2.1 in (Nesterov, 2003). Remark that the subgradient method can achieve a $\mathcal{O}(1/t)$ rate when $f$ is strongly convex. We leave the analysis of TOS for strongly convex nonsmooth $f$ as an open problem.

## 4 TOS for Stochastic Setting

In this section, we focus on the three-composite stochastic optimization template:

$$\min_{x \in \mathbb{R}^n} \quad \phi(x) := f(x) + g(x) + h(x) \quad \text{where} \quad f(x) := \mathbb{E}_\xi \tilde{f}(x, \xi) \tag{14}$$

and $\xi$ is a random variable. The following theorem characterizes the convergence rate of Algorithm 1 for Problem (14).

**Theorem 2.** *Consider Problem (14) and employ TOS (Algorithm 1) with a fixed step-size $\gamma_t = \gamma = \gamma_0/\sqrt{T+1}$ for some $\gamma_0 > 0$. Suppose we are receiving the update directions $u_t$ from an unbiased stochastic first-order oracle with bounded variance, i.e.,*

$$\hat{u}_t := \mathbb{E}[u_t | z_t] \in \partial f(z_t) \quad \text{and} \quad \mathbb{E}[\|u_t - \hat{u}_t\|^2] \le \sigma^2 \text{ for some } \sigma < +\infty. \tag{15}$$

*Assume that $\|\hat{u}_t\| \le G_f$ for all $t$. Then, the following guarantees hold:*

$$\mathbb{E}[f(\bar{z}_T) + g(\bar{z}_T) + h(\bar{x}_T)] - \phi_\star \le \frac{1}{2\sqrt{T+1}} \left( \frac{D^2}{\gamma_0} + \gamma_0 (\sigma^2 + G_f^2) \right) \quad \text{and} \tag{16}$$

$$\mathbb{E}[\|\bar{x}_T - \bar{z}_T\|] \le \frac{2}{T+1} \left( D + \gamma_0 \left( G_f + \frac{\sigma}{2} \right) \right), \quad \text{where} \;\; D = \max\{\|y_0 - x_\star\|, \|y_0 - y_\star\|\}. \tag{17}$$

**Remark 3.** *Similar rate guarantees hold with some restrictions on the choice of $\gamma_0$ if we replace bounded subgradients assumption with the smoothness of $f$. We defer details to the supplements.*

If we can estimate $D, G_f$ and $\sigma$, then we can optimize the bounds by choosing $\gamma_0 \approx D/\max\{G_f, \sigma\}$. This gives $f(\bar{z}_T) + g(\bar{z}_T) + h(\bar{x}_T) - \phi_\star \le \mathcal{O}(D\max\{G_f, \sigma\}/\sqrt{T})$ and $\|\bar{x}_T - \bar{z}_T\| \le \mathcal{O}(D/T)$.

Analogous to Corollary 1, from Theorem 2 we can derive convergence guarantees when $h$ is Lipschitz continuous or an indicator function. As in the nonsmooth setting, the rates shown in this section are optimal because Problem (14) covers $g(x) = h(x) = 0$ as a special case.

## 5 TOS with Adaptive Learning Rates

In this section, we focus on an important subclass of Problem (1) where $g$ and $h$ are indicator functions of some closed and convex sets:

$$\min_{x \in \mathbb{R}^n} \quad f(x) \quad \text{subject to} \quad x \in \mathcal{G} \cap \mathcal{H}. \tag{18}$$

TOS is effective for Problem (18) when projections onto $\mathcal{G}$ and $\mathcal{H}$ are easy but the projection onto their intersection is challenging. Particular examples include transportation polytopes, doubly nonnegative matrices, and isotonic regression, among many others.

We propose ADAPTOS with an adaptive step-size in the spirit of adaptive online learning algorithms and online to batch conversion techniques, see (Duchi et al., 2011; Rakhlin & Sridharan, 2013; Levy, 2017; Levy et al., 2018; Cutkosky, 2019; Kavis et al., 2019; Bach & Levy, 2019) and the references therein. ADAPTOS employs the following step-size rule:

$$\gamma_t = \frac{\alpha}{\sqrt{\beta + \sum_{\tau=0}^{t-1} \|u_\tau\|^2}} \quad \text{for some } \alpha, \beta > 0. \tag{19}$$

$\beta$ in the denominator prevents $\gamma_t$ to become undefined. If $D := \|y_0 - x_\star\|$ and $G_f$ are known, theory suggests choosing $\alpha = D$ and $\beta = G_f^2$ for a tight upper bound, however, this choice affects only the constants and not the rate of convergence as we demonstrate in the rest of this section. Importantly, we do not assume any prior knowledge on $D$ or $G_f$. In practice, we often discard $\beta$ and use $\gamma_0 = \alpha$ at the first iteration.

For ADAPTOS, in addition to (5), we will also use a second ergodic sequence with weighted averaging:

$$\tilde{x}_t = \frac{1}{\sum_{\tau=0}^t \gamma_\tau} \sum_{\tau=0}^t \gamma_\tau x_\tau \quad \text{and} \quad \tilde{z}_t = \frac{1}{\sum_{\tau=0}^t \gamma_\tau} \sum_{\tau=0}^t \gamma_\tau z_\tau. \tag{20}$$

This sequence was also considered for TOS with line-search in (Pedregosa & Gidel, 2018).

**Theorem 3.** *Consider Problem* (18) *and TOS (Algorithm 1) with the update directions $u_t \in \partial f(z_t)$ and the adaptive step-size* (19). *Assume that $\|u_t\| \leq G_f$ for all $t$. Then, the estimates generated by TOS satisfy*

$$f(\tilde{z}_t) - f_\star \leq \tilde{\mathcal{O}}\left(\frac{2\alpha G_f}{\sqrt{t+1}}\left(\frac{D^2}{4\alpha^2} + 1 + \frac{G_f}{\sqrt{\beta}}\right)\right) \quad and \tag{21}$$

$$\mathrm{dist}(\bar{z}_t, \mathcal{H}) \leq \tilde{\mathcal{O}}\left(\frac{2\alpha}{\sqrt{t+1}}\left(1 + \frac{G_f}{\sqrt{\beta}}\right)\right) \quad where \quad D = \|y_0 - x_\star\|. \tag{22}$$

If $D$ and $G_f$ are known, we can choose $\alpha = D$ and $\beta = G_f^2$. This gives $f(\tilde{z}_t) - f_\star \leq \tilde{\mathcal{O}}(G_f D/\sqrt{t})$ and $\mathrm{dist}(\bar{z}_t, \mathcal{H}) \leq \tilde{\mathcal{O}}(D/\sqrt{t})$.

The next theorem establishes a faster rate for the same algorithm when $f$ is smooth and a solution lies in the interior of the feasible set.

**Theorem 4.** *Consider Problem* (18) *and suppose $f$ is $L_f$-smooth on $\mathcal{G}$. Use TOS (Algorithm 1) with the update directions $u_t = \nabla f(z_t)$ and the adaptive step-size* (19). *Assume that $\|u_t\| \leq G_f$ for all $t$. Suppose Problem* (18) *has a solution in the interior of the feasible set. Then, the estimates generated by TOS satisfy*

$$f(\bar{z}_t) - f_\star \leq \tilde{\mathcal{O}}\left(\frac{2}{t+1}\left(4\alpha^2 L_f\left(\frac{D^2}{4\alpha^2} + 1 + \frac{G_f^2}{\beta}\right)^2 + \alpha\sqrt{\beta}\left(\frac{D^2}{4\alpha^2} + 1 + \frac{G_f^2}{\beta}\right)\right)\right) \quad and \tag{23}$$

$$\mathrm{dist}(\bar{z}_t, \mathcal{H}) \leq \tilde{\mathcal{O}}\left(\frac{2\alpha}{t+1}\left(\frac{D}{\alpha} + 1 + \frac{G_f}{\sqrt{\beta}}\right)\right) \quad where \quad D = \|y_0 - x_\star\|. \tag{24}$$

If $D$ and $G_f$ are known, we can choose $\alpha = D$ and $\beta = G_f^2$.
This gives $f(\bar{z}_t) - f_\star \leq \tilde{\mathcal{O}}((L_f D^2 + G_f D)/t)$ and $\mathrm{dist}(\bar{z}_t, \mathcal{H}) \leq \tilde{\mathcal{O}}(D/t)$.

**Remark 4.** *When $f$ is smooth, the boundedness assumption $\|u_t\| \leq G_f$ holds automatically with $G_f \leq L_f D_\mathcal{G}$ if $\mathcal{G}$ has a bounded diameter $D_\mathcal{G}$.*

We believe the assumption on the location of the solution is a limitation of the analysis and that the method can achieve fast rates when $f$ is smooth regardless of where the solution lies. Remark that this assumption also appears in (Levy, 2017; Levy et al., 2018).

Following the definition in (Nesterov, 2015), we say that an algorithm is universal if it does not require to know whether the objective is smooth or not yet it implicitly adapts to the smoothness of the objective. ADAPTOS attains universal convergence rates for Problem (18). It converges to a solution with $\tilde{\mathcal{O}}(1/\sqrt{t})$ rate (in function value) when $f$ is nonsmooth. The rate becomes $\tilde{\mathcal{O}}(1/t)$ if $f$ is smooth and the solution is in the interior of the feasible set.

Finally, the next theorem shows that ADAPTOS can successfully handle stochastic (sub)gradients.

**Theorem 5.** *Consider Problem* (18). *Use TOS (Algorithm 1) with the update directions $u_t$ from an unbiased stochastic subgradient oracle such that $\mathbb{E}[u_t|z_t] \in \partial f(z_t)$ almost surely. Assume that $\|u_t\| \leq G_f$ for all $t$. Suppose Problem* (18) *has a solution in the interior of the feasible set. Then, the estimates generated by TOS satisfy*

$$\mathbb{E}\big[f(\tilde{z}_t) - f_\star\big] \leq \tilde{\mathcal{O}}\left(\frac{2\alpha G_f}{\sqrt{t+1}}\left(\frac{D^2}{4\alpha^2} + 1 + \frac{G_f^2}{\beta}\right)\right) \quad and \tag{25}$$

$$\mathbb{E}[\mathrm{dist}(\bar{z}_t, \mathcal{H})] \leq \tilde{\mathcal{O}}\left(\frac{2\alpha}{t+1}\left(\frac{D}{\alpha} + 1 + \frac{G_f}{\sqrt{\beta}}\right)\right) \quad where \quad D = \|y_0 - x_\star\|. \tag{26}$$

## 6   Numerical Experiments

This section demonstrates empirical performance of the proposed method on a number of convex optimization problems. We also present an experiment on neural networks. Our experiments are performed in Python 3.7 with Intel Core i9-9820X CPU @ 3.30GHz. We present more experiments on isotonic regression and portfolio optimization in the supplementary materials. The source code for the experiments is available in the supplements.

## 6.1 Experiments on Convex Optimization with Smooth $f$

In this subsection, we compare ADAPTOS with TOS, PDHG and their line-search variants TOS-LS and PDHG-LS. Our experiments are based on the benchmarks described in (Pedregosa & Gidel, 2018) and their source code available in COPT Library (Pedregosa et al., 2020) under the new BSD License. We implement ADAPTOS and investigate its performance on three different problems:

▷ Logistic regression with *overlapping group lasso* penalty:

$$\min_{x \in \mathbb{R}^n} \quad \frac{1}{N} \sum_{i=1}^N \log(1 + \exp(-b_i \langle a_i, x \rangle)) + \lambda \sum_{G \in \mathcal{G}} \sqrt{|G|} \|x_G\| + \lambda \sum_{H \in \mathcal{H}} \sqrt{|H|} \|x_H\|, \quad (27)$$

where $\{(a_1, b_1), \ldots, (a_N, b_N)\}$ is a given set of training examples, $\mathcal{G}$ and $\mathcal{H}$ are the sets of distinct groups and $|\cdot|$ denotes the cardinality. The model we use (from COPT) considers groups of size 10 with 2 overlapping coefficients. In this experiment, we use the benchmarks on synthetic data (dimensions $n = 1002$, $N = 100$) and real-sim dataset (Chang & Lin, 2011) ($n = 20958$, $N = 72309$).

▷ Image recovery with *total variation* penalty:

$$\min_{X \in \mathbb{R}^{m \times n}} \quad \|Y - \mathcal{A}(X)\|_F^2 + \lambda \sum_{i=1}^m \sum_{j=1}^{n-1} |X_{i,j+1} - X_{i,j}| + \lambda \sum_{j=1}^n \sum_{i=1}^{m-1} |X_{i+1,j} - X_{i,j}|, \quad (28)$$

where $Y$ is a given blurred image and $\mathcal{A} : \mathbb{R}^{m \times n} \to \mathbb{R}^{m \times n}$ is a linear operator (blur kernel). The benchmark in COPT solves this problem for an image of size $153 \times 115$ with a provided blur kernel.

▷ Sparse and low-rank matrix recovery via $\ell_1$ *and nuclear-norm* regularizations:

$$\min_{X \in \mathbb{R}^{n \times n}} \quad \frac{1}{N} \sum_{i=1}^N \texttt{huber}(b_i - \langle A_i, X \rangle) + \lambda \|X\|_* + \lambda \|X\|_1. \quad (29)$$

We use `huber` loss. $\{(A_1, b_1), \ldots, (A_N, b_N)\}$ is a given set of measurements and $\|X\|_1$ is the vector $\ell_1$-norm of $X$. The benchmark in COPT considers a symmetric ground truth matrix $X^\natural \in \mathbb{R}^{20 \times 20}$ and noisy synthetic measurements ($N = 100$) where $A_i$ has Gaussian iid entries. $b_i = \langle A_i, X^\natural \rangle + \omega_i$ where $\omega_i$ is generated from a zero-mean unit variance Gaussian distribution.

At each problem, we consider two different values for the regularization parameter $\lambda$. We use all methods with their default parameters in the benchmark. For ADAPTOS, we discard $\beta$ and tune $\alpha$ by trying the powers of 10. See the supplementary material for the behavior of the algorithm with different values of $\alpha$. Figure 1 shows the results of this experiment. In most cases, the performance of ADAPTOS is between TOS-LS and PDHG-LS. Remark that TOS-LS is using the extra knowledge of the Lipschitz constant of $h$.

## 6.2 Experiments on Convex Optimization with Nonsmooth $f$

We examine the empirical performance of ADAPTOS for nonsmooth problems on an image impainting and denoising task from (Zeng & So, 2018; Yurtsever et al., 2018). We are given an occluded image (*i.e.*, missing some pixels) of size $517 \times 493$, contaminated with salt and pepper noise of $10\%$ density. We use the following template where data fitting is measured in terms of vector $\ell_p$-norm:

$$\min_{\boldsymbol{X} \in \mathbb{R}^{m \times n}} \quad \|\mathcal{A}(X) - Y\|_p \quad \text{subject to} \quad \|X\|_* \leq \lambda, \quad 0 \leq X \leq 1, \quad (30)$$

where $Y$ is the observed noisy image with missing pixels. This is essentially a matrix completion problem, $\mathcal{A} : \mathbb{R}^{m \times n} \to \mathbb{R}^{m \times n}$ is a linear map that samples the observed pixels in $Y$. In particular, we consider (30) with $p = 1$ and $p = 2$. The $\ell_2$-loss is common in practice for matrix completion (often in the least-squares form) but it is not robust against the outliers induced by the salt and pepper noise. $\ell_1$-loss is known to be more reliable for this task.

The subgradients in both cases have a fixed norm at all points (note that the subgradients are binary valued for $\ell_1$-loss and unit-norm for $\ell_2$-loss), hence the analytical and the adaptive step-sizes are same up to a constant factor.

Figure 2 shows the results. The empirical rates for $p = 1$ roughly match our guarantees in Theorem 1. We observe a locally linear convergence rate when $\ell_2$-loss is used. Interestingly, the ergodic sequence converges faster than the last iterate for $p = 1$ but significantly slower for $p = 2$. The runtime of the two settings are approximately the same, with 67 msec per iteration on average. Despite the slower rates, we found $\ell_1$-loss more practical on this problem. A low-accuracy solution obtained by 1000 iterations on $\ell_1$-loss yields a high quality recovery with PSNR 26.21 dB, whereas the PSNR saturates at 21.15 dB for the $\ell_2$-formulation. See the supplements for the recovered images and more details.

### 6.3 An Experiment on Neural Networks

In this section, we train a regularized deep neural network to test our methods on nonconvex optimization. We consider a regularized neural network problem formulation in (Scardapane et al., 2017). This problem involves a fully connected neural network with the standard cross-entropy loss function, a ReLu activation for the hidden layers, and the softmax activation for the output layer. Two regularizers are added to this loss function: The first one is the standard $\ell_1$ regularizer, and the second is the group sparse regularizer where the outgoing connections of each neuron is considered as a group. The goal is to force all outgoing connections from the same neurons to be simultaneously zero, so that we can safely remove the neurons from the network. This is shown as an effective way to obtain compact networks (Scardapane et al., 2017), which is crucial for the deployment of the learned parameters on resource-constrained devices such as smartphones (Blalock et al., 2020).

We reuse the open source implementation (built with Lasagne framework based on Theano) published in (Scardapane et al., 2017) under BSD-2 License. We follow their experimental setup and instructions with MNIST database (LeCun, 1998) containing 70k grayscale images ($28 \times 28$) of handwritten digits (split 75/25 into train and test partitions). We train a fully connected neural network with 784 input features, three hidden layers ($400/300/100$) and 10-dimensional output layer. Interested readers can find more details on the implementation in the supplementary material or in (Scardapane et al., 2017).

Scardapane et al. (2017) use SGD and Adam with the subgradient of the overall objective. In contrast, our methods can leverage the prox-operators for the regularizers. Figure 3 compares the performance in terms of two measures: the sparsity of the parameters and the accuracy. On the left side, we see the spectrum of weight and neuron magnitudes. The advantage of using prox-operators is outstanding: More than 93% of the weights are zero and 68% of neurons are inactive when trained with ADAPTOS. In contrast, subgradient based methods can achieve only approximately sparse solutions.

The third and the fourth subplots present the training and test accuracies. Remarkably, ADAPTOS performs better than the state-of-the-art (both in train and test). Unfortunately, we could not achieve the same performance gain in preliminary experiments with more complex models like ResNet (He et al., 2016), where SGD with momentum shines. Interested readers can find the code for these preliminary experiments in the supplements. We leave the technical analysis and a comprehensive examination of ADAPTOS for nonconvex problems to a future work.

## 7 Conclusions

We studied an extension of TOS that permits subgradients and stochastic gradients instead of the gradient step and established convergence guarantees for this extension. Moreover, we proposed an adaptive step-size rule (ADAPTOS) for the minimization of a convex function over the intersection of two convex sets. ADAPTOS guarantees a nearly optimal $\tilde{\mathcal{O}}(1/\sqrt{t})$ rate on the baseline setting, and it enjoys the faster $\tilde{\mathcal{O}}(1/t)$ rate when the problem is smooth and the solution is in the interior of feasible set. We present numerical experiments on various benchmark problems. The empirical performance of the method is promising.

We conclude with a short list of open questions and follow-up directions: (i) In parallel to the subgradient method, we believe TOS can achieve $\mathcal{O}(1/t)$ rate guarantees in the nonsmooth setting if $f$ is strongly convex. The analysis remains open. (ii) The faster rate for ADAPTOS on smooth $f$ requires an extra assumption on the location of the solution. We believe this assumption can be removed, and leave this as an open problem. (iii) We analyzed ADAPTOS only for a specific subclass of Problem (1) in which $g$ and $h$ are indicator functions. Extending this result for the whole class is a valuable question for future study.

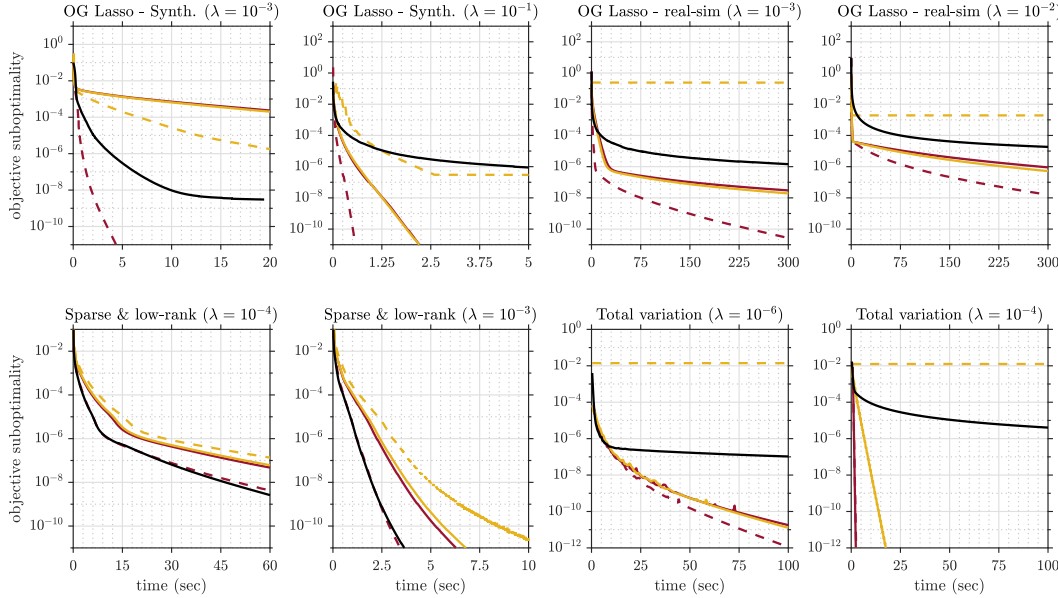

Figure 1: Empirical comparison of 5 algorithms for Problem (1) with smooth $f$. Dashed lines represent the line-search variants of TOS and PDHG. The performance of ADAPTOS is between TOS-LS and PDHG-LS. TOS and PDHG require the knowledge of the smoothness constant, and TOS-LS uses the Lipschitz constant for one of the nonsmooth terms.

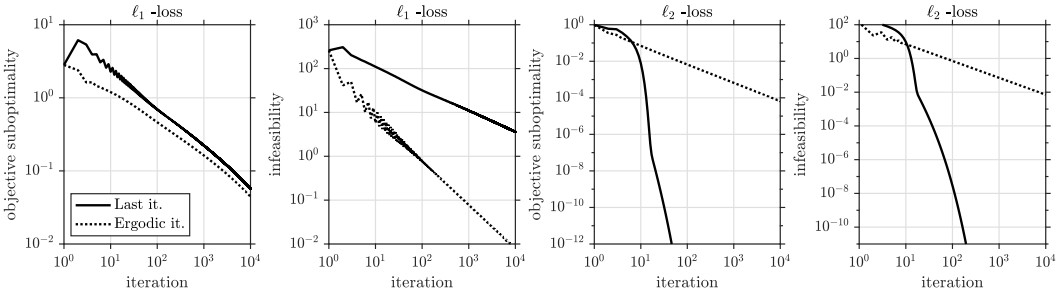

Figure 2: Performance of ADAPTOS on image impainting and denoising problems with $\ell_1$ and $\ell_2$-loss functions. The empirical rates for $\ell_1$-loss match the guaranteed $\mathcal{O}(1/\sqrt{t})$ rate in objective suboptimality and $\mathcal{O}(1/t)$ in infeasibility. We observe a locally linear convergence rate for the $\ell_2$-loss.

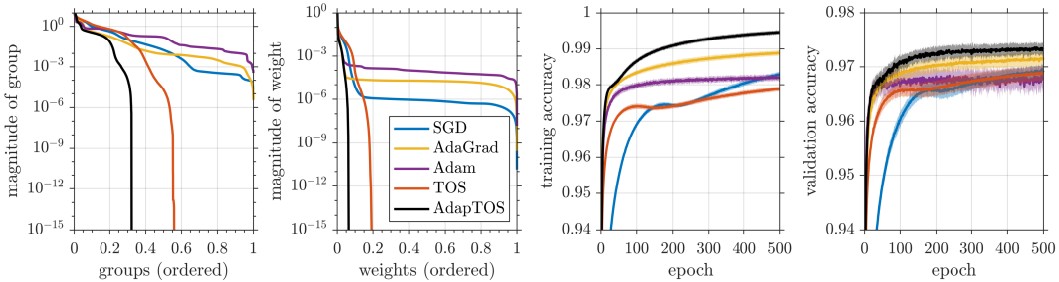

Figure 3: Comparison of methods on training neural networks with group lasso regularization. The outgoing connections of each neuron form a group. The first plot shows the magnitude of weights after 500 epochs. The second plot shows the absolute sum of outgoing weights from each neuron. x-axes are normalized by the total number of weights and neurons in these plots. More than 68% of the neurons are inactive on the network trained by ADAPTOS. The third and fourth plots show the training and validation losses. This experiment is performed with 20 random seeds. The solid lines show the average performance and the shaded area represents $\pm$ standard deviation from the mean.

## Acknowledgements

The authors would like to thank Ahmet Alacaoglu for carefully reading and reporting an error in the preprint of this paper.

Alp Yurtsever received support from the Swiss National Science Foundation Early Postdoc.Mobility Fellowship P2ELP2_187955, from the Wallenberg AI, Autonomous Systems and Software Program (WASP) funded by the Knut and Alice Wallenberg Foundation, and partial postdoctoral support from the NSF-CAREER grant IIS-1846088. Suvrit Sra acknowledges support from an NSF BIGDATA grant (1741341) and an NSF CAREER grant (1846088).

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
