## Supplementary Material

## A  Preliminaries

We will use the following standard results in our analysis.

**Lemma S.2.** *Let $f : \mathbb{R}^n \to \mathbb{R} \cup \{+\infty\}$ be a proper closed and convex function. Then, for any $x, u \in \mathbb{R}^n$, the followings are equivalent:*

*(i)* $u = \text{prox}_f(x)$.

*(ii)* $x - u \in \partial f(u)$.

*(iii)* $\langle x - u, y - u \rangle \leq f(y) - f(u)$ *for any* $y \in \mathbb{R}^n$.

**Corollary S.2** (Firm non-expansivity of the prox-operator)**.** *Let $f : \mathbb{R}^n \to \mathbb{R} \cup \{+\infty\}$ be a proper closed and convex function. Then, for any $x, u \in \mathbb{R}^n$, the followings hold:*

$$\text{(non-expansivity)} \qquad \|\text{prox}_f(x) - \text{prox}_f(y)\| \leq \|x - y\|.$$

$$\text{(firm non-expansivity)} \qquad \|\text{prox}_f(x) - \text{prox}_f(y)\|^2 \leq \langle x - y, \text{prox}_f(x) - \text{prox}_f(y) \rangle.$$

## B  Fixed Point Characterization

This appendix presents the proof for Lemma 1. This is a straightforward extension of Lemma 2.2 in (Davis & Yin, 2017) to permit subgradients. We will use this lemma in the next section to prove the boundedness of $y_t$ in Algorithm 1.

### B.1  Proof of Lemma 1

Define $z = \text{prox}_{\gamma g}(y)$ and $x = \text{prox}_{\gamma h}(2z - y - \gamma u)$. Then, $\text{TOS}_\gamma(y, u) := y - z + x$.

Suppose there exists $u \in \partial f(z)$ such that $\text{TOS}(y, u) = y$. Then, we must have $z = x$. Moreover, by Lemma S.2, we have

$$z = \text{prox}_{\gamma g}(y) \iff y - z \in \gamma \partial g(z), \tag{S.1}$$
$$\text{and} \quad z = x = \text{prox}_{\gamma h}(2z - y - \gamma u) \iff z - y - \gamma u \in \gamma \partial h(x). \tag{S.2}$$

By summing up (S.1) and (S.2), we observe

$$0 \in \gamma(u + \partial g(z) + \partial h(x)) \implies 0 \in \partial f(z) + \partial g(z) + \partial h(z) = \partial \phi(z), \tag{S.3}$$

which proves that $z$ is an optimal solution of Problem (1) since $\phi$ is convex.

To prove the reverse direction, suppose $z$ is an optimal solution, i.e., there exists $u \in \partial f(z), v \in \partial g(z), w \in \partial h(z)$ such that $u + v + w = 0$. By Lemma S.2, we have

$$z = \text{prox}_{\gamma g}(y) \iff y - z \in \gamma \partial g(z), \tag{S.4}$$
$$\text{and} \quad x = \text{prox}_{\gamma h}(2z - y - \gamma u) \iff 2z - x - y - \gamma u \in \gamma \partial h(x). \tag{S.5}$$

Now, let $y = z + \gamma v$. Then,

$$2z - x - y - \gamma u = z - x - \gamma(u + v) = z - x + \gamma w. \tag{S.6}$$

Therefore, we have $z - x + \gamma w \in \partial h(x)$. Again, due to Lemma S.2, this means $x = \text{prox}_{\gamma h}(z + \gamma w)$. We also know $w \in \partial h(z) \iff z + \gamma w - z \in \partial \gamma h(z) \iff z = \text{prox}_{\gamma h}(z + \gamma w)$. However, since $h$ is convex, its prox-operator is unique, hence, $x = z$ and $\text{TOS}_\gamma(y, u) = y$.

## C   Boundedness Guarantees

**Theorem S.6.** *Consider Problem (1) and employ TOS (Algorithm 1) with subgradient steps $u_t \in \partial f(z_t)$ and a fixed step-size $\gamma = \gamma_0/\sqrt{T+1}$ for some $\gamma_0 > 0$. Assume that $\|u_t\| \leq G_f$ for all t. Then,*

$$\|y_{T+1} - y_\star\| \leq \|y_0 - y_\star\| + 2\gamma_0 G_f \tag{S.7}$$

*where $y_\star$ is a fixed point of TOS.*

*Proof.* By Lemma 1, there exists $u_\star \in \partial f(x_\star)$ such that

$$x_\star = \mathrm{prox}_{\gamma g}(y_\star) = \mathrm{prox}_{\gamma h}(2x_\star - y_\star - \gamma u_\star) = z_\star. \tag{S.8}$$

We decompose $\|y_{t+1} - y_\star\|^2$ as

$$\|y_{t+1} - y_\star\|^2 = \|y_t - z_t + x_t - y_\star + x_\star - x_\star\|^2$$
$$= \|y_t - z_t - y_\star + x_\star\|^2 + \|x_t - x_\star\|^2 + 2\langle x_t - x_\star, y_t - z_t - y_\star + x_\star\rangle. \tag{S.9}$$

Since $z_t = \mathrm{prox}_{\gamma g}(y_t)$ and $x_\star = \mathrm{prox}_{\gamma g}(y_\star)$, by the firm non-expansivity of the prox-operator, we have

$$\|y_t - z_t - y_\star + x_\star\|^2 = \langle y_t - z_t - y_\star + x_\star, y_t - y_\star\rangle - \langle y_t - z_t - y_\star + x_\star, z_t - x_\star\rangle$$
$$\leq \langle y_t - z_t - y_\star + x_\star, y_t - y_\star\rangle. \tag{S.10}$$

Similarly, since $x_t = \mathrm{prox}_{\gamma h}(2z_t - y_t - \gamma u_t)$ and $x_\star = \mathrm{prox}_{\gamma h}(2x_\star - y_\star - \gamma u_\star)$, by the firm non-expansivity of the prox-operator, we have

$$\|x_t - x_\star\|^2 \leq \langle x_t - x_\star, (2z_t - y_t - \gamma u_t) - (2x_\star - y_\star - \gamma u_\star)\rangle. \tag{S.11}$$

By combining (S.9), (S.10) and (S.11), we get

$$\|y_{t+1} - y_\star\|^2 \leq \langle y_t - z_t + x_t - y_\star, y_t - y_\star\rangle - \gamma\langle x_t - x_\star, u_t - u_\star\rangle$$
$$= \langle y_{t+1} - y_\star, y_t - y_\star\rangle - \gamma\langle x_t - x_\star, u_t - u_\star\rangle$$
$$= \frac{1}{2}\|y_{t+1} - y_\star\|^2 + \frac{1}{2}\|y_t - y_\star\|^2 - \frac{1}{2}\|y_{t+1} - y_t\|^2 - \gamma\langle x_t - x_\star, u_t - u_\star\rangle. \tag{S.12}$$

Since $u_t \in \partial f(z_t)$ and $u_\star \in \partial f(x_\star)$, we have

$$-\langle x_t - x_\star, u_t - u_\star\rangle = -\langle z_t - x_\star, u_t - u_\star\rangle - \langle x_t - z_t, u_t - u_\star\rangle$$
$$\leq -\langle x_t - z_t, u_t - u_\star\rangle$$
$$\leq \frac{1}{2\gamma}\|x_t - z_t\|^2 + \frac{\gamma}{2}\|u_t - u_\star\|^2, \tag{S.13}$$

where we used Young's inequality in the last line. We use this inequality in (S.12) to obtain

$$\|y_{t+1} - y_\star\|^2 \leq \|y_t - y_\star\|^2 + \gamma^2\|u_t - u_\star\|^2. \tag{S.14}$$

If we sum this inequality from $t = 0$ to $T$, we get

$$\|y_{T+1} - y_\star\|^2 \leq \|y_0 - y_\star\|^2 + \gamma^2\sum_{\tau=0}^{T}\|u_\tau - u_\star\|^2. \tag{S.15}$$

Finally, due to the bounded subgradients assumption, we have $\|u_\tau - u_\star\| \leq 2G_f$, hence

$$\|y_{T+1} - y_\star\|^2 \leq \|y_0 - y_\star\|^2 + 4G_f^2\gamma^2(T+1) = \|y_0 - y_\star\|^2 + 4G_f^2\gamma_0^2. \tag{S.16}$$

We complete the proof by taking the square-root of both sides,

$$\|y_{T+1} - y_\star\| \leq \sqrt{\|y_0 - y_\star\|^2 + 4G_f^2\gamma_0^2} \leq \|y_0 - y_\star\| + 2G_f\gamma_0. \tag{S.17}$$

$\square$

**Theorem S.7.** *Consider Problem (14) and employ TOS (Algorithm 1) with a fixed step-size $\gamma = \gamma_0/\sqrt{T+1}$ for some $\gamma_0 > 0$. Suppose we are receiving the update directions $u_t$ from an unbiased stochastic first-order oracle with bounded variance such that*

$$\hat{u}_t := \mathbb{E}[u_t|z_t] \in \partial f(z_t) \quad and \quad \mathbb{E}[\|u_t - \hat{u}_t\|^2] \leq \sigma^2 \text{ for some } \sigma < +\infty. \tag{S.18}$$

*Assume that $\|\hat{u}_t\| \leq G_f$ for all t. Then,*

$$\mathbb{E}[\|y_{T+1} - y_\star\|] \leq \|y_0 - y_\star\| + \gamma_0(2G_f + \sigma) \tag{S.19}$$

*where $y_\star$ is a fixed point of TOS.*

*Proof.* We follow the same steps as in the proof of Theorem S.6 until (S.12):

$$\|y_{t+1} - y_\star\|^2 \leq \|y_t - y_\star\|^2 - \|y_{t+1} - y_t\|^2 - 2\gamma\langle x_t - x_\star, u_t - u_\star\rangle. \tag{S.20}$$

Then, we need to take noise into account:

$$-\langle x_t - x_\star, u_t - u_\star\rangle = -\langle x_t - z_t, u_t - u_\star\rangle - \langle z_t - x_\star, u_t - \hat{u}_t\rangle - \langle z_t - x_\star, \hat{u}_t - u_\star\rangle$$
$$\leq -\langle x_t - z_t, u_t - u_\star\rangle - \langle z_t - x_\star, u_t - \hat{u}_t\rangle. \tag{S.21}$$

We take the expectation of both sides and get

$$-\mathbb{E}[\langle x_t - x_\star, u_t - u_\star\rangle] \leq -\mathbb{E}[\langle x_t - z_t, u_t - u_\star\rangle]$$
$$\leq \frac{1}{2\gamma}\mathbb{E}[\|x_t - z_t\|^2] + \frac{\gamma}{2}\mathbb{E}[\|u_\star - u_t\|^2]$$
$$= \frac{1}{2\gamma}\mathbb{E}[\|x_t - z_t\|^2] + \frac{\gamma}{2}\mathbb{E}[\|u_\star - \hat{u}_t\|^2] + \frac{\gamma}{2}\mathbb{E}[\|\hat{u}_t - u_t\|^2]$$
$$\leq \frac{1}{2\gamma}\mathbb{E}[\|x_t - z_t\|^2] + 2\gamma G_f^2 + \frac{\gamma}{2}\sigma^2, \tag{S.22}$$

where the last line holds due to the bounded subgradients and variance assumptions.

Now, we take the expectation of (S.20) and substitute (S.22) into it:

$$\mathbb{E}[\|y_{t+1} - y_\star\|^2] \leq \mathbb{E}[\|y_t - y_\star\|^2] + \gamma^2(4G_f^2 + \sigma^2). \tag{S.23}$$

Finally, we sum this inequality over $t = 0, 1, \ldots, T$:

$$\mathbb{E}[\|y_{T+1} - y_\star\|^2] \leq \|y_0 - y_\star\|^2 + \gamma_0^2(4G_f^2 + \sigma^2). \tag{S.24}$$

By Jensen's inequality, we have $\mathbb{E}[\|y_{T+1} - y_\star\|]^2 \leq \mathbb{E}[\|y_{T+1} - y_\star\|^2]$. We finalize the proof by taking the square-root of both sides. □

Next, we assume that $f$ is $L_f$-smooth instead of Lipschitz continuity.

**Theorem S.8.** *Consider Problem* (14) *and suppose $f$ is $L_f$-smooth on* $\mathrm{dom}(g)$. *Employ TOS (Algorithm 1) with a fixed step-size $\gamma_t = \gamma = \gamma_0/\sqrt{T+1}$ for some $\gamma_0 \in [0, \frac{2}{L_f}]$. Suppose we are receiving the update directions $u_t$ from an unbiased stochastic first-order oracle with bounded variance such that*

$$\mathbb{E}[u_t|z_t] = \nabla f(z_t) \quad and \quad \mathbb{E}[\|u_t - \nabla f(z_t)\|^2] \leq \sigma^2 \ for \ some \ \sigma < +\infty. \tag{S.25}$$

*Then,*

$$\mathbb{E}[\|y_{T+1} - y_\star\|] \leq \|y_0 - y_\star\| + 2\sigma\sqrt{\frac{\gamma_0}{L_f}}, \tag{S.26}$$

*where $y_\star$ is a fixed point of TOS.*

*Proof.* The proof is similar to the proof of Theorem S.7. We start from (S.20) and take the expectation:

$$\mathbb{E}[\|y_{t+1} - y_\star\|^2] \leq \mathbb{E}[\|y_t - y_\star\|^2] - \mathbb{E}[\|y_{t+1} - y_t\|^2] - 2\gamma\mathbb{E}[\langle x_t - x_\star, u_t - u_\star\rangle]. \tag{S.27}$$

We decompose the last term as follows:

$$\mathbb{E}[\langle x_t - x_\star, u_\star - u_t\rangle] = \mathbb{E}[\langle x_t - z_t, u_\star - u_t\rangle] + \mathbb{E}[\langle z_t - x_\star, u_\star - \nabla f(z_t)\rangle] + \mathbb{E}[\langle z_t - x_\star, \nabla f(z_t) - u_t\rangle]$$
$$\leq \mathbb{E}[\langle x_t - z_t, u_\star - u_t\rangle] - \frac{1}{L_f}\mathbb{E}[\|u_\star - \nabla f(z_t)\|^2]. \tag{S.28}$$

where the inequality holds since $f$ is $L_f$-smooth and convex. Moreover, we can bound the inner product term by using Young's inequality as follows:

$$\mathbb{E}[\langle x_t - z_t, u_\star - u_t\rangle] = \mathbb{E}\big[\langle x_t - z_t, u_\star - \nabla f(z_t)\rangle + \langle x_t - z_t, \nabla f(z_t) - u_t\rangle\big]$$
$$\leq \mathbb{E}\Big[\frac{c_1 + c_2}{2}\|x_t - z_t\|^2 + \frac{1}{2c_1}\|u_\star - \nabla f(z_t)\|^2 + \frac{1}{2c_2}\|\nabla f(z_t) - u_t\|^2\Big] \tag{S.29}$$

for any $c_1, c_2 > 0$. We choose $c_1 = L_f/2$, so that the corresponding terms in (S.28) and (S.29) cancel out. Combining (S.27), (S.28) and (S.29), we get

$$\mathbb{E}[\|y_{t+1} - y_\star\|^2] \leq \mathbb{E}[\|y_t - y_\star\|^2] - \mathbb{E}[\|y_{t+1} - y_t\|^2] + \gamma(\tfrac{L_f}{2} + c_2)\mathbb{E}[\|x_t - z_t\|^2] + \gamma\frac{\sigma^2}{c_2}$$

$$\leq \mathbb{E}[\|y_t - y_\star\|^2] + \left(\gamma\frac{L_f + 2c_2}{2} - 1\right)\mathbb{E}[\|x_t - z_t\|^2] + \gamma\frac{\sigma^2}{c_2}. \tag{S.30}$$

Then, we choose $c_2 = \frac{L_f}{2}(\sqrt{T+1} - 1)$. With the condition $\gamma_0 \leq \frac{2}{L_f}$, this guarantees

$$\gamma\frac{L_f + 2c_2}{2} - 1 = \frac{\gamma_0}{\sqrt{T+1}}\frac{L_f\sqrt{T+1}}{2} - 1 \leq \gamma_0\frac{L_f}{2} - 1 \leq 0. \tag{S.31}$$

Returning to (S.30), we now have

$$\mathbb{E}[\|y_{t+1} - y_\star\|^2] \leq \mathbb{E}[\|y_t - y_\star\|^2] + \frac{\gamma_0}{\sqrt{T+1}}\frac{2\sigma^2}{L_f(\sqrt{T+1}-1)} \leq \mathbb{E}[\|y_t - y_\star\|^2] + \frac{4\gamma_0\sigma^2}{L_f(T+1)}. \tag{S.32}$$

Finally, we sum this inequality over $t = 0$ to $T$,

$$\mathbb{E}[\|y_{T+1} - y_\star\|^2] \leq \|y_0 - y_\star\|^2 + \frac{4\gamma_0\sigma^2}{L_f}. \tag{S.33}$$

Remark that $\mathbb{E}[\|y_{T+1} - y_\star\|]^2 \leq \mathbb{E}[\|y_{T+1} - y_\star\|^2]$. We finish the proof by taking the square-root of both sides. $\square$

## D  Convergence Guarantees

This section presents the technical analysis of our main results.

### D.1  Proof of Theorem 1

We divide this proof into two parts.

**Part 1.** In the first part, we show that the sequence generated by TOS satisfies

$$\langle u_t, x_t - x_\star\rangle + g(z_t) - g(x_\star) + h(x_t) - h(x_\star) \leq \frac{1}{2\gamma}\|y_t - x_\star\|^2 - \frac{1}{2\gamma}\|y_{t+1} - x_\star\|^2 - \frac{1}{2\gamma}\|y_{t+1} - y_t\|^2. \tag{S.34}$$

Since $x_t = \text{prox}_{\gamma h}(2z_t - y_t - \gamma u_t)$, by Lemma S.2, we have

$$\langle 2z_t - y_t - \gamma u_t - x_t, x_\star - x_t\rangle \leq \gamma h(x_\star) - \gamma h(x_t). \tag{S.35}$$

We rearrange this inequality as follows:

$$\langle u_t, x_t - x_\star\rangle + h(x_t) - h(x_\star) \leq \frac{1}{\gamma}\langle 2z_t - y_t - x_t, x_t - x_\star\rangle$$

$$= \frac{1}{\gamma}\langle z_t - y_t, z_t - x_\star\rangle + \frac{1}{\gamma}\langle z_t - y_t, x_t - z_t\rangle + \frac{1}{\gamma}\langle z_t - x_t, x_t - x_\star\rangle$$

$$= \frac{1}{\gamma}\langle z_t - y_t, z_t - x_\star\rangle + \frac{1}{\gamma}\langle y_t + x_t - z_t - x_\star, z_t - x_t\rangle$$

$$= \frac{1}{\gamma}\langle z_t - y_t, z_t - x_\star\rangle + \frac{1}{\gamma}\langle y_{t+1} - x_\star, y_t - y_{t+1}\rangle. \tag{S.36}$$

Then, we use Lemma S.2 once again (for $\gamma g$) and get

$$\langle u_t, x_t - x_\star\rangle + g(z_t) - g(x_\star) + h(x_t) - h(x_\star) \leq \frac{1}{\gamma}\langle y_{t+1} - x_\star, y_t - y_{t+1}\rangle$$

$$\leq \frac{1}{2\gamma}\|y_t - x_\star\|^2 - \frac{1}{2\gamma}\|y_{t+1} - x_\star\|^2 - \frac{1}{2\gamma}\|y_{t+1} - y_t\|^2. \tag{S.37}$$

This completes the first part of the proof.

S.4

**Part 2.** In the second part, we characterize the convergence rate of $f(\bar{z}_t) + g(\bar{z}_t) + h(\bar{x}_t) - \phi_\star$ to 0 by using (S.34). Since $f$ is convex, we have

$$\langle u_t, x_t - x_\star \rangle = \langle u_t, z_t - x_\star \rangle - \langle u_t, z_t - x_t \rangle \geq f(z_t) - f(x_\star) - \frac{1}{2\gamma} \|x_t - z_t\|^2 - \frac{\gamma}{2} \|u_t\|^2. \tag{S.38}$$

By combining (S.34) and (S.38), we obtain

$$f(z_t) + g(z_t) + h(x_t) - \phi_\star \leq \frac{1}{2\gamma} \|y_t - x_\star\|^2 - \frac{1}{2\gamma} \|y_{t+1} - x_\star\|^2 + \frac{\gamma}{2} \|u_t\|^2. \tag{S.39}$$

We sum this inequality over $t = 0$ to $T$:

$$\sum_{\tau=0}^{T} \left( f(z_\tau) + g(z_\tau) + h(x_\tau) - \phi_\star \right) \leq \frac{1}{2\gamma} \|y_0 - x_\star\|^2 + \frac{\gamma}{2} \sum_{\tau=0}^{T} \|u_\tau\|^2 \leq \frac{1}{2\gamma} \|y_0 - x_\star\|^2 + \frac{\gamma_0}{2} G_f^2 \sqrt{T+1}, \tag{S.40}$$

where the second inequality holds due to the bounded subgradients assumption. Finally, we divide both sides by $(T+1)$ and use Jensen's inequality:

$$f(\bar{z}_t) + g(\bar{z}_t) + h(\bar{x}_t) - \phi_\star \leq \frac{1}{2\sqrt{T+1}} \left( \frac{1}{\gamma_0} \|y_0 - x_\star\|^2 + \gamma_0 G_f^2 \right). \tag{S.41}$$

## D.2 Proof of Theorem 2

The proof is similar to the proof of Theorem 1. We will only discuss the different steps. Part 1 of the proof is the same, *i.e.,* (S.34) is still valid.

We need to consider the randomness of the gradient estimator in the second part. To this end, we modify (S.38) as:

$$\begin{aligned}
\mathbb{E}[\langle u_t, x_t - x_\star \rangle] &= \mathbb{E}[\langle \hat{u}_t, z_t - x_\star \rangle] + \mathbb{E}[\langle u_t - \hat{u}_t, z_t - x_\star \rangle] - \mathbb{E}[\langle u_t, z_t - x_t \rangle] \\
&\geq \mathbb{E}[f(z_t) - f(x_\star)] - \mathbb{E}[\langle u_t, z_t - x_t \rangle] \\
&\geq \mathbb{E}[f(z_t) - f(x_\star)] - \frac{1}{2\gamma} \mathbb{E}[\|z_t - x_t\|^2] - \frac{\gamma}{2} \mathbb{E}[\|u_t\|^2] \\
&\geq \mathbb{E}[f(z_t) - f(x_\star)] - \frac{1}{2\gamma} \mathbb{E}[\|z_t - x_t\|^2] - \frac{\gamma}{2} (G_f^2 + \sigma^2),
\end{aligned} \tag{S.42}$$

where the last line holds since

$$\mathbb{E}[\|u_t\|^2] = \mathbb{E}[\|u_t - \hat{u}_t + \hat{u}_t\|^2] \tag{S.43}$$

$$= \mathbb{E}[\|u_t - \hat{u}_t\|^2] + \mathbb{E}[\|\hat{u}_t\|^2] + 2\mathbb{E}[\langle u_t - \hat{u}_t, \hat{u}_t \rangle] \leq \sigma^2 + G_f^2. \tag{S.44}$$

Now, we take the expectation of (S.34) and substitute (S.42) into it:

$$\mathbb{E}[f(z_t) + g(z_t) + h(x_t)] - \phi_\star \leq \frac{1}{2\gamma} \mathbb{E}[\|y_t - x_\star\|^2] - \frac{1}{2\gamma} \mathbb{E}[\|y_{t+1} - x_\star\|^2] + \frac{\gamma}{2}(\sigma^2 + G_f^2). \tag{S.45}$$

We sum this inequality from $t = 0$ to $T$ and divide both sides by $T + 1$. Then, we use Jensen's inequality and get

$$\mathbb{E}[f(\bar{z}_T) + g(\bar{z}_T) + h(\bar{x}_T)] - \phi_\star \leq \frac{1}{2\sqrt{T+1}} \left( \frac{1}{\gamma_0} \|y_0 - x_\star\|^2 + \gamma_0(\sigma^2 + G_f^2) \right). \tag{S.46}$$

## D.3 TOS for the Smooth and Stochastic Setting (Remark 3)

**Theorem S.9.** *Consider Problem* (14) *and suppose* $f$ *is* $L_f$-*smooth on* $\mathrm{dom}(g)$. *Employ TOS (Algorithm 1) with a fixed step-size* $\gamma = \gamma_0 / \sqrt{T+1}$ *for some* $\gamma_0 \in [0, \frac{1}{2L_f}]$. *Suppose we are receiving the update directions* $u_t$ *from an unbiased stochastic first-order oracle with bounded variance such that*

$$\mathbb{E}[u_t | z_t] = \nabla f(z_t) \quad \text{and} \quad \mathbb{E}[\|u_t - \nabla f(z_t)\|^2] \leq \sigma^2 \text{ for some } \sigma < +\infty. \tag{S.47}$$

*Then, the following guarantees hold:*

$$\mathbb{E}[f(\bar{x}_T) + g(\bar{z}_T) + h(\bar{x}_T)] - \phi_\star \leq \frac{1}{\sqrt{T+1}} \left( \frac{D^2}{2\gamma_0} + \gamma_0 \sigma^2 \right), \quad \text{where} \quad D = \|y_0 - x_\star\|. \tag{S.48}$$

S.5

*Proof.* The proof is similar to Theorem 1. (S.34) still holds. We modify (S.38) as follows (similar to (S.42)):

$$\mathbb{E}[\langle u_t, x_t - x_\star \rangle] \geq \mathbb{E}[f(z_t) - f(x_\star)] + \mathbb{E}[\langle \nabla f(z_t) - u_t, z_t - x_t \rangle] - \mathbb{E}[\langle \nabla f(z_t), z_t - x_t \rangle]$$

$$\geq \mathbb{E}[f(x_t) - f(x_\star)] - \frac{1}{4\gamma}\mathbb{E}[\|z_t - x_t\|^2] - \gamma\mathbb{E}[\|\nabla f(z_t) - u_t\|^2] - \frac{L_f}{2}\|x_t - z_t\|^2$$

$$\geq \mathbb{E}[f(x_t) - f(x_\star)] - \frac{1 + 2\gamma L_f}{4\gamma}\mathbb{E}[\|y_{t+1} - y_t\|^2] - \gamma\sigma^2. \tag{S.49}$$

We take the expectation of (S.34) and replace (S.49) into it

$$\mathbb{E}[f(x_t) + g(z_t) + h(x_t)] - \phi_\star \leq \frac{1}{2\gamma}\|y_t - x_\star\|^2 - \frac{1}{2\gamma}\|y_{t+1} - x_\star\|^2 + \frac{2\gamma L_f - 1}{4\gamma}\mathbb{E}[\|y_{t+1} - y_t\|^2] + \gamma\sigma^2$$

$$\leq \frac{1}{2\gamma}\|y_t - x_\star\|^2 - \frac{1}{2\gamma}\|y_{t+1} - x_\star\|^2 + \gamma\sigma^2, \tag{S.50}$$

where the second line holds since we choose $\gamma_0 \in [0, \frac{1}{2L_f}]$.

We sum (S.50) from $t = 0$ to $T$ and divide both sides by $T + 1$. We complete the proof by using Jensen's inequality. □

# E Convergence Guarantees for ADAPTOS

In this section, we focus on Problem (18), an important subclass of Problem (1) where $g$ and $h$ are indicator functions. In this setting, TOS performs the following steps iteratively for $t = 0, 1, \ldots$:

$$z_t = \text{proj}_{\mathcal{G}}(y_t) \tag{S.51}$$

$$x_t = \text{proj}_{\mathcal{H}}(2z_t - y_t - \gamma_t u_t) \tag{S.52}$$

$$y_{t+1} = y_t - z_t + x_t, \tag{S.53}$$

where $\gamma_t$ at line (S.52) is chosen according to the adaptive step-size rule (19), *i.e.*,

$$\gamma_t = \frac{\alpha}{\sqrt{\beta + \sum_{\tau=0}^{t-1}\|u_\tau\|^2}} \qquad \text{for some } \alpha, \beta > 0. \tag{S.54}$$

The following lemmas are useful in the analysis.

**Lemma S.3** (Lemma A.2 in (Levy, 2017)). *Let $f : \mathbb{R}^n \to \mathbb{R}$ be a $L_f$-smooth function and let $x_\star \in \arg\min_{x \in \mathbb{R}^n} f(x)$. Then,*

$$\|\nabla f(x)\|^2 \leq 2L_f\big(f(x) - f(x_\star)\big), \qquad \forall x \in \mathbb{R}^n.$$

**Lemma S.4** (Lemma 9 in (Bach & Levy, 2019)). *For any non-negative numbers $a_0, \ldots, a_t \in [0, a]$, and $\beta \geq 0$*

$$\sum_{i=0}^{t} \frac{a_i}{\sqrt{\beta + \sum_{j=0}^{i-1} a_j}} \leq \frac{2a}{\sqrt{\beta}} + 3\sqrt{a} + 3\sqrt{\beta + \sum_{i=0}^{t-1} a_i}.$$

**Lemma S.5** (Lemma 10 in (Bach & Levy, 2019)). *For any non-negative numbers $a_0, \ldots, a_t \in [0, a]$, and $\beta \geq 0$*

$$\sum_{i=0}^{t} \frac{a_i}{\beta + \sum_{j=0}^{i-1} a_j} \leq 2 + \frac{4a}{\beta} + 2\log\left(1 + \sum_{i=0}^{t-1} \frac{a_i}{\beta}\right).$$

**Corollary S.3.** *Suppose $\|u_t\| \leq G$ for all $t$. Then, the following relations hold for* ADAPTOS*:*

$(i).\ \sum_{\tau=0}^{t} \gamma_\tau \|u_\tau\|^2 = \alpha \sum_{\tau=0}^{t} \frac{\|u_\tau\|^2}{\sqrt{\beta + \sum_{j=0}^{\tau-1}\|u_j\|^2}} \leq \alpha\left(\frac{2G^2}{\sqrt{\beta}} + 3G + 3\sqrt{\beta + G^2 t}\right)$

$(ii).\ \sum_{\tau=0}^{t} \gamma_\tau^2 \|u_\tau\|^2 = \alpha^2 \sum_{\tau=0}^{t} \frac{\|u_\tau\|^2}{\beta + \sum_{j=0}^{\tau-1}\|u_j\|^2} \leq \alpha^2\left(2 + \frac{4G^2}{\beta} + 2\log\left(1 + \frac{G^2}{\beta}t\right)\right)$

$(iii).\ \sum_{\tau=0}^{t} \gamma_\tau \|u_\tau\| = \sum_{\tau=0}^{t} \sqrt{\gamma_\tau^2\|u_\tau\|^2} \leq \sqrt{(t+1)\sum_{\tau=0}^{t}\gamma_\tau^2\|u_\tau\|^2} \leq \alpha\sqrt{t+1}\sqrt{2 + \frac{4G^2}{\beta} + 2\log\left(1 + \frac{G^2}{\beta}t\right)}$

## E.1   Proof of Theorem 3

First, we will bound the growth rate of $\|y_{t+1} - x_\star\|$. We decompose $\|y_{t+1} - x_\star\|^2$ as

$$\|y_{t+1} - x_\star\|^2 = \|y_t - z_t + x_t - x_\star\|^2 = \|y_t - z_t\|^2 + \|x_t - x_\star\|^2 + 2\langle x_t - x_\star, y_t - z_t\rangle. \tag{S.55}$$

Since $z_t = \text{proj}_{\mathcal{G}}(y_t)$ and $x_\star \in \mathcal{G}$, we have

$$\|y_t - z_t\|^2 = \langle y_t - z_t, y_t - x_\star\rangle + \langle y_t - z_t, x_\star - z_t\rangle \le \langle y_t - z_t, y_t - x_\star\rangle. \tag{S.56}$$

Similarly, since $x_t = \text{proj}_{\mathcal{H}}(2z_t - y_t - \gamma_t u_t)$ and $x_\star \in \mathcal{H}$, by the firm non-expansivity, we have

$$\|x_t - x_\star\|^2 \le \langle x_t - x_\star, 2z_t - y_t - \gamma_t u_t - x_\star\rangle. \tag{S.57}$$

By combining (S.55), (S.56) and (S.57), we get

$$\begin{aligned}
\|y_{t+1} - x_\star\|^2 &\le \langle y_t - z_t + x_t - x_\star, y_t - x_\star\rangle - \gamma_t\langle x_t - x_\star, u_t\rangle \\
&= \langle y_{t+1} - x_\star, y_t - x_\star\rangle - \gamma_t\langle x_t - x_\star, u_t\rangle \\
&= \frac{1}{2}\|y_{t+1} - x_\star\|^2 + \frac{1}{2}\|y_t - x_\star\|^2 - \frac{1}{2}\|y_{t+1} - y_t\|^2 - \gamma_t\langle x_t - x_\star, u_t\rangle.
\end{aligned} \tag{S.58}$$

Now, we rearrange (S.58) as follows:

$$\begin{aligned}
\|y_{t+1} - x_\star\|^2 &\le \|y_t - x_\star\|^2 - \|y_{t+1} - y_t\|^2 + 2\gamma_t\langle u_t, x_\star - x_t\rangle \\
&= \|y_t - x_\star\|^2 - \|y_{t+1} - y_t\|^2 + 2\gamma_t\langle u_t, x_\star - z_t\rangle + 2\gamma_t\langle u_t, z_t - x_t\rangle \\
&\le \|y_t - x_\star\|^2 + 2\gamma_t\langle u_t, x_\star - z_t\rangle + \gamma_t^2\|u_t\|^2 \\
&\le \|y_t - x_\star\|^2 + 2\gamma_t\|u_t\|\|z_t - x_\star\| + \gamma_t^2\|u_t\|^2 \\
&\le \|y_t - x_\star\|^2 + 2\gamma_t\|u_t\|\|y_t - x_\star\| + \gamma_t^2\|u_t\|^2 = (\|y_t - x_\star\| + \gamma_t\|u_t\|)^2,
\end{aligned} \tag{S.59}$$

where we use non-expansivity of the projection operator in the last line: $\|z_t - x_\star\| = \|\text{proj}_{\mathcal{G}}(y_t) - x_\star\| \le \|y_t - x_\star\|$.

Next, we take the square root of both sides and use Corollary S.3 to get

$$\begin{aligned}
\|y_{t+1} - x_\star\| &\le \|y_t - x_\star\| + \gamma_t\|u_t\| \\
&\le \|y_0 - x_\star\| + \sum_{\tau=0}^{t}\gamma_\tau\|u_\tau\| \\
&\le \|y_0 - x_\star\| + \alpha\sqrt{t+1}\sqrt{2 + \frac{4G_f^2}{\beta} + 2\log\left(1 + \frac{G_f^2}{\beta}t\right)}.
\end{aligned} \tag{S.60}$$

Now, we can derive a bound on the infeasibility as follows:

$$\text{dist}(\bar{z}_t, \mathcal{H}) \le \|\bar{x}_t - \bar{z}_t\| = \frac{1}{t+1}\|\sum_{\tau=0}^{t}(x_\tau - z_\tau)\| = \frac{1}{t+1}\|y_{t+1} - y_0\| \le \frac{1}{t+1}(\|y_{t+1} - x_\star\| + \|y_0 - x_\star\|)$$

$$\le \frac{1}{t+1}\left(2\|y_0 - x_\star\| + \alpha\sqrt{t+1}\sqrt{2 + \frac{4G_f^2}{\beta} + 2\log\left(1 + \frac{G_f^2}{\beta}t\right)}\right). \tag{S.61}$$

Next, we prove convergence in objective value. Define $s_t = \sum_{\tau=0}^{t}\gamma_t$ and $\tilde{z}_t = \frac{1}{s_t}\sum_{\tau=0}^{t}\gamma_t z_t$. Since $f$ is convex, by Jensen's inequality,

$$f(\tilde{z}_t) - f_\star \le \frac{1}{s_t}\sum_{\tau=0}^{t}\gamma_\tau\left(f(z_\tau) - f_\star\right) \le \frac{1}{s_t}\sum_{\tau=0}^{t}\gamma_\tau\langle u_t, z_\tau - x_\star\rangle. \tag{S.62}$$

From (S.59), we have

$$\gamma_t\langle u_t, z_t - x_\star\rangle \le \frac{1}{2}\|y_t - x_\star\|^2 - \frac{1}{2}\|y_{t+1} - x_\star\|^2 + \frac{1}{2}\gamma_t^2\|u_t\|^2. \tag{S.63}$$

If we substitute (S.63) into (S.62), we obtain

$$\begin{aligned}
f(\tilde{z}_t) - f_\star &\le \frac{1}{2s_t}\left(\|y_0 - x_\star\|^2 + \sum_{\tau=0}^{t}\gamma_\tau^2\|u_\tau\|^2\right) \\
&\le \frac{1}{2s_t}\left(\|y_0 - x_\star\|^2 + \alpha^2\left(2 + \frac{4G_f^2}{\beta} + 2\log\left(1 + \frac{G_f^2}{\beta}t\right)\right)\right)
\end{aligned} \tag{S.64}$$

where the second line comes from Corollary S.3. Finally, we note that

$$s_t = \sum_{\tau=0}^{t} \gamma_\tau = \sum_{\tau=0}^{t} \frac{\alpha}{\sqrt{\beta + \sum_{j=0}^{\tau-1} \|u_j\|^2}} \geq \sum_{\tau=0}^{t} \frac{\alpha}{\sqrt{\beta + G_f^2 t}} = \frac{\alpha(t+1)}{\sqrt{\beta + G_f^2 t}} \geq \frac{\alpha(t+1)}{\sqrt{\beta} + G_f \sqrt{t}}. \tag{S.65}$$

We complete the proof by using (S.65) in (S.64):

$$f(\tilde{z}_t) - f_\star \leq \left( \frac{G_f}{\sqrt{t+1}} + \frac{\sqrt{\beta}}{t+1} \right) \left( \frac{1}{2\alpha} \|y_0 - x_\star\|^2 + \alpha \left( 1 + \frac{2G_f^2}{\beta} + \log\left(1 + \frac{G_f^2}{\beta} t\right) \right) \right). \tag{S.66}$$

## E.2  Proof of Theorem 4

As in the proof of Theorem 3, our first goal is to bound $\|y_{t+1} - y_\star\|$. We start from (S.59):

$$\|y_{t+1} - x_\star\|^2 \leq \|y_t - x_\star\|^2 + 2\gamma_t \langle u_t, x_\star - z_t \rangle + \gamma_t^2 \|u_t\|^2. \tag{S.67}$$

By assumption $f$ is convex and the solution lies in the interior of the feasible set. Hence, $\langle u_t, x_\star - z_t \rangle \leq 0$ and

$$\|y_{t+1} - x_\star\|^2 \leq \|y_t - x_\star\|^2 + \gamma_t^2 \|u_t\|^2 \leq \|y_0 - x_\star\|^2 + \sum_{\tau=0}^{t} \gamma_\tau^2 \|u_\tau\|^2. \tag{S.68}$$

By using Corollary S.3, this leads to

$$\|y_{t+1} - x_\star\|^2 \leq \underbrace{\|y_0 - x_\star\|^2 + \alpha^2 \left( 2 + \frac{4G_f^2}{\beta} + 2\log\left(1 + \frac{G_f^2}{\beta} t\right) \right)}_{:=D_t^2}. \tag{S.69}$$

We take the square-root of both sides to obtain $\|y_{t+1} - x_\star\| \leq D_t$. This proves that $\|y_t - x_\star\|$ is bounded by a logarithmic growth. Similar to (S.61), we can use this bound to prove convergence to a feasible point:

$$\text{dist}(\bar{z}_t, \mathcal{H}) \leq \|\bar{x}_t - \bar{z}_t\| \leq \frac{1}{t+1}\left( \|y_{t+1} - x_\star\| + \|y_0 - x_\star\| \right) \leq \frac{1}{t+1}\left( D_t + \|y_0 - x_\star\| \right)$$

$$\leq \frac{1}{t+1}\left( 2\|y_0 - x_\star\| + \alpha\sqrt{2 + \frac{4G_f^2}{\beta} + 2\log\left(1 + \frac{G_f^2}{\beta} t\right)} \right). \tag{S.70}$$

Next, we analyze the objective suboptimality. From (S.67), we have

$$\langle u_t, z_t - x_\star \rangle \leq \frac{1}{2\gamma_t} \|y_t - x_\star\|^2 - \frac{1}{2\gamma_t} \|y_{t+1} - x_\star\|^2 + \frac{\gamma_t}{2} \|u_t\|^2. \tag{S.71}$$

Then, since $f$ is convex, by using Jensen's inequality and (S.71), we get

$$\Phi_t := \frac{1}{t+1} \sum_{\tau=0}^{t} (f(z_\tau) - f_\star)$$

$$\leq \frac{1}{t+1} \sum_{\tau=0}^{t} \langle u_t, z_t - x_\star \rangle$$

$$\leq \frac{1}{2(t+1)} \left( \frac{1}{\gamma_0} \|y_0 - x_\star\|^2 + \underbrace{\sum_{\tau=1}^{t} \left( \frac{1}{\gamma_\tau} - \frac{1}{\gamma_{\tau-1}} \right) \|y_\tau - x_\star\|^2 + \sum_{\tau=0}^{t} \gamma_\tau \|u_\tau\|^2}_{(*)} \right). \tag{S.72}$$

S.8

Now, we focus on $(*)$. By using (S.68), we get

$$(*) \leq \sum_{\tau=1}^{t} \left( \frac{1}{\gamma_\tau} - \frac{1}{\gamma_{\tau-1}} \right) \left( \|y_0 - x_\star\|^2 + \sum_{j=0}^{\tau-1} \gamma_j^2 \|u_j\|^2 \right)$$

$$= \left( \frac{1}{\gamma_t} - \frac{1}{\gamma_0} \right) \|y_0 - x_\star\|^2 + \sum_{\tau=1}^{t} \frac{1}{\gamma_\tau} \sum_{j=0}^{\tau-1} \gamma_j^2 \|u_j\|^2 - \sum_{\tau=1}^{t} \frac{1}{\gamma_{\tau-1}} \sum_{j=0}^{\tau-1} \gamma_j^2 \|u_j\|^2$$

$$= \left( \frac{1}{\gamma_t} - \frac{1}{\gamma_0} \right) \|y_0 - x_\star\|^2 + \sum_{\tau=1}^{t} \frac{1}{\gamma_\tau} \sum_{j=0}^{\tau-1} \gamma_j^2 \|u_j\|^2 - \sum_{\tau=0}^{t-1} \frac{1}{\gamma_\tau} \sum_{j=0}^{\tau} \gamma_j^2 \|u_j\|^2$$

$$= \left( \frac{1}{\gamma_t} - \frac{1}{\gamma_0} \right) \|y_0 - x_\star\|^2 + \sum_{\tau=1}^{t} \frac{1}{\gamma_\tau} \sum_{j=0}^{\tau} \gamma_j^2 \|u_j\|^2 - \sum_{\tau=1}^{t} \gamma_\tau \|u_\tau\|^2 - \sum_{\tau=0}^{t-1} \frac{1}{\gamma_\tau} \sum_{j=0}^{\tau} \gamma_j^2 \|u_j\|^2$$

$$= \left( \frac{1}{\gamma_t} - \frac{1}{\gamma_0} \right) \|y_0 - x_\star\|^2 + \sum_{\tau=0}^{t} \frac{1}{\gamma_\tau} \sum_{j=0}^{\tau} \gamma_j^2 \|u_j\|^2 - \sum_{\tau=0}^{t} \gamma_\tau \|u_\tau\|^2 - \sum_{\tau=0}^{t-1} \frac{1}{\gamma_\tau} \sum_{j=0}^{\tau} \gamma_j^2 \|u_j\|^2$$

$$= \left( \frac{1}{\gamma_t} - \frac{1}{\gamma_0} \right) \|y_0 - x_\star\|^2 + \frac{1}{\gamma_t} \sum_{j=0}^{t} \gamma_j^2 \|u_j\|^2 - \sum_{\tau=0}^{t} \gamma_\tau \|u_\tau\|^2. \tag{S.73}$$

We substitute this back into (S.72) and obtain

$$\Phi_t \leq \frac{1}{2(t+1)} \frac{1}{\gamma_t} \left( \|y_0 - x_\star\|^2 + \sum_{j=0}^{t} \gamma_j^2 \|u_j\|^2 \right) \leq \frac{D_t^2}{2\gamma_t(t+1)} \tag{S.74}$$

where $D_t^2$ is defined in (S.69).

By the definition of $\gamma_t$ we get

$$\Phi_t \leq \frac{D_t^2}{2\gamma_t(t+1)} = \frac{D_t^2}{2\alpha(t+1)} \sqrt{\beta + \sum_{\tau=0}^{t-1} \|u_\tau\|^2} \leq \frac{D_t^2}{2\alpha(t+1)} \sqrt{\beta + \sum_{\tau=0}^{t} \|u_\tau\|^2}. \tag{S.75}$$

By Lemma S.3, we have

$$\sum_{\tau=0}^{t} \|u_\tau\|^2 \leq 2L_f \sum_{\tau=0}^{t} \left( f(z_\tau) - f_\star \right) = 2L_f(t+1)\Phi_t. \tag{S.76}$$

We place this back into (S.75), take the square of both sides, and rearrange the inequality as follows:

$$\frac{4\alpha^2(t+1)^2}{D_t^4} \Phi_t^2 \leq 2L_f(t+1)\Phi_t + \beta. \tag{S.77}$$

This is a second order inequality of $\Phi_t$. By solving this inequality, we get

$$\Phi_t \leq \frac{1}{2(t+1)} \left( \left( \frac{D_t^2}{\alpha} \right)^2 L_f + \frac{D_t^2}{\alpha} \sqrt{\beta} \right). \tag{S.78}$$

Finally, we note $f(\bar{z}_t) - f_\star \leq \Phi_t$ by Jensen's inequality.

### E.3 Proof of Theorem 5

Once again we start from (S.59) and take the expectation of both sides:

$$\mathbb{E}[\|y_{t+1} - x_\star\|^2] \leq \mathbb{E}[\|y_t - x_\star\|^2] + \mathbb{E}[2\gamma_t \langle u_t, x_\star - z_t \rangle] + \mathbb{E}[\gamma_t^2 \|u_t\|^2]$$

$$\leq \mathbb{E}[\|y_t - x_\star\|^2] + \mathbb{E}[2\gamma_t \langle \hat{u}_t, x_\star - z_t \rangle] + \mathbb{E}[\gamma_t^2 \|u_t\|^2]. \tag{S.79}$$

Since we assume $f$ is convex and the solution lies in the interior of the feasible set, we know $0 \in \partial f(x_\star)$ and $\langle \hat{u}_t, x_\star - z_t \rangle \leq 0$. Hence, we have

$$\mathbb{E}[\|y_{t+1} - x_\star\|^2] \leq \mathbb{E}[\|y_t - x_\star\|^2] + \mathbb{E}[\gamma_t^2 \|u_t\|^2] \leq \|y_0 - x_\star\|^2 + \mathbb{E}\left[ \sum_{\tau=0}^{t} \gamma_\tau^2 \|u_\tau\|^2 \right]. \tag{S.80}$$

By using Corollary S.3, this leads to

$$\mathbb{E}[\|y_{t+1} - x_\star\|^2] \le \|y_0 - x_\star\|^2 + \alpha^2 \left(2 + \tfrac{4G_f^2}{\beta} + 2\log\left(1 + \tfrac{G_f^2}{\beta}t\right)\right). \tag{S.81}$$

We take the square-root of both sides. Note that $\mathbb{E}[\|y_{t+1} - x_\star\|]^2 \le \mathbb{E}[\|y_{t+1} - x_\star\|^2]$, hence

$$\mathbb{E}[\|y_{t+1} - x_\star\|] \le \|y_0 - x_\star\| + \alpha\sqrt{2 + \tfrac{4G_f^2}{\beta} + 2\log\left(1 + \tfrac{G_f^2}{\beta}t\right)}. \tag{S.82}$$

Similar to (S.61), we can use this bound to prove convergence to a feasible point:

$$\mathbb{E}[\mathrm{dist}(\bar{z}_t, \mathcal{H})] \le \frac{1}{t+1}\left(\mathbb{E}[\|y_{t+1} - x_\star\|] + \|y_0 - x_\star\|\right)$$

$$\le \frac{1}{t+1}\left(2\|y_0 - x_\star\| + \alpha\sqrt{2 + \tfrac{4G_f^2}{\beta} + 2\log\left(1 + \tfrac{G_f^2}{\beta}t\right)}\right). \tag{S.83}$$

Next, we analyze convergence in the function value. Note that

$$\gamma_\tau(f(z_\tau) - f_\star) \le \gamma_\tau\langle \hat{u}_t, z_\tau - x_\star\rangle = \gamma_\tau\langle u_\tau, z_\tau - x_\star\rangle + \gamma_\tau\langle \hat{u}_\tau - u_\tau, z_\tau - x_\star\rangle. \tag{S.84}$$

Recall that $\gamma_\tau$ and $u_\tau$ are independent given $z_\tau$. Then, the second term vanishes if we take the expectation of both sides:

$$\mathbb{E}[\gamma_\tau(f(z_\tau) - f_\star)] = \mathbb{E}[\gamma_\tau\langle u_\tau, z_\tau - x_\star\rangle]. \tag{S.85}$$

Then, by using (S.79), we get

$$\mathbb{E}[\gamma_\tau(f(z_\tau) - f_\star)] \le \frac{1}{2}\mathbb{E}\left[\|y_\tau - x_\star\|^2 - \|y_{\tau+1} - x_\star\|^2 + \gamma_\tau^2\|u_\tau\|^2\right]. \tag{S.86}$$

If we sum this inequality over $\tau = 0, 1, \ldots, t$, we get

$$\mathbb{E}\left[\sum_{\tau=0}^{t}\gamma_\tau(f(z_\tau) - f_\star)\right] \le \frac{1}{2}\|y_0 - x_\star\|^2 + \frac{1}{2}\mathbb{E}\left[\sum_{\tau=0}^{t}\gamma_\tau^2\|u_\tau\|^2\right]. \tag{S.87}$$

From Corollary S.3,

$$\mathbb{E}\left[\sum_{\tau=0}^{t}\gamma_\tau^2\|u_\tau\|^2\right] \le \alpha^2\left(2 + \tfrac{4G_f^2}{\beta} + 2\log\left(1 + \tfrac{G_f^2}{\beta}t\right)\right). \tag{S.88}$$

Replacing this back into (S.87), we get

$$\mathbb{E}\left[\sum_{\tau=0}^{t}\gamma_\tau(f(z_\tau) - f_\star)\right] \le \frac{1}{2}\|y_0 - x_\star\|^2 + \alpha^2\left(1 + \tfrac{2G_f^2}{\beta} + \log\left(1 + \tfrac{G_f^2}{\beta}t\right)\right). \tag{S.89}$$

Let us define $s_t := \sum_{\tau=0}^{t}\gamma_\tau$. By Jensen's inequality, we get

$$\mathbb{E}\left[\sum_{\tau=0}^{t}\gamma_\tau(f(z_\tau) - f_\star)\right] = \mathbb{E}\left[\frac{s_t}{s_t}\sum_{\tau=0}^{t}\gamma_\tau(f(z_\tau) - f_\star)\right] \ge \mathbb{E}[s_t(f(\tilde{z}_t) - f_\star)]. \tag{S.90}$$

Note that

$$s_t = \sum_{\tau=0}^{t}\gamma_\tau = \sum_{\tau=0}^{t}\frac{\alpha}{\sqrt{\beta + \sum_{j=0}^{\tau-1}\|u_j\|^2}} \ge \sum_{\tau=0}^{t}\frac{\alpha}{\sqrt{\beta + G_f^2 t}} = \frac{\alpha(t+1)}{\sqrt{\beta + G_f^2 t}}. \tag{S.91}$$

Then, we have

$$\frac{\alpha(t+1)}{\sqrt{\beta + G_f^2 t}}\mathbb{E}[(f(\tilde{z}_t) - f_\star)] \le \mathbb{E}[s_t(f(\tilde{z}_t) - f_\star)] \le \frac{1}{2}\|y_0 - x_\star\|^2 + \alpha^2\left(1 + \tfrac{2G_f^2}{\beta} + \log\left(1 + \tfrac{G_f^2}{\beta}t\right)\right). \tag{S.92}$$

By rearranging, we get

$$\mathbb{E}[(f(\tilde{z}_t) - f_\star)] \le \left(\frac{\sqrt{\beta}}{t+1} + \frac{G_f}{\sqrt{t+1}}\right)\left(\frac{1}{2\alpha}\|y_0 - x_\star\|^2 + \alpha\left(1 + \tfrac{2G_f^2}{\beta} + \log\left(1 + \tfrac{G_f^2}{\beta}t\right)\right)\right). \tag{S.93}$$

# F   More Details on the Experiments in Section 6

## F.1   Details for Section 6.1

In the implementation of ADAPTOS we simply discarded $\beta$ and set $\gamma_0 = \alpha$. Figure S.5 demonstrates how the performance of ADAPTOS depends on $\alpha$ for the experiments we considered in Section 6.1.

For Figure 1, we choose:

▷ $\alpha = 10$ for overlapping group lasso with $\lambda = 10^{-3}$ and synthetic data,

▷ $\alpha = 1$ for overlapping group lasso with $\lambda = 10^{-1}$ and synthetic data,

▷ $\alpha = 100$ for overlapping group lasso with $\lambda = 10^{-3}$ and real-sim dataset,

▷ $\alpha = 100$ for overlapping group lasso with $\lambda = 10^{-2}$ and real-sim dataset,

▷ $\alpha = 1$ for sparse and low-rank regularization with $\lambda = 10^{-3}$,

▷ $\alpha = 1$ for sparse and low-rank regularization with $\lambda = 10^{-4}$,

▷ $\alpha = 100$ for total variation deblurring with $\lambda = 10^{-6}$,

▷ $\alpha = 100$ for total variation deblurring with $\lambda = 10^{-4}$.

In Section 6.1, we consider problems only with smooth $f$. To present how the performance of ADAPTOS changes by $\alpha$ when $f$ is nonsmooth, we also run the overlapping group lasso problem with the hinge loss. In this setting, we used RCV1 dataset (Lewis et al., 2004) ($n = 677399$, $N = 20242$) and tried two different values of the regularization parameter $\lambda = 10^{-3}$ and $10^{-2}$. The results are shown in Figure S.4.

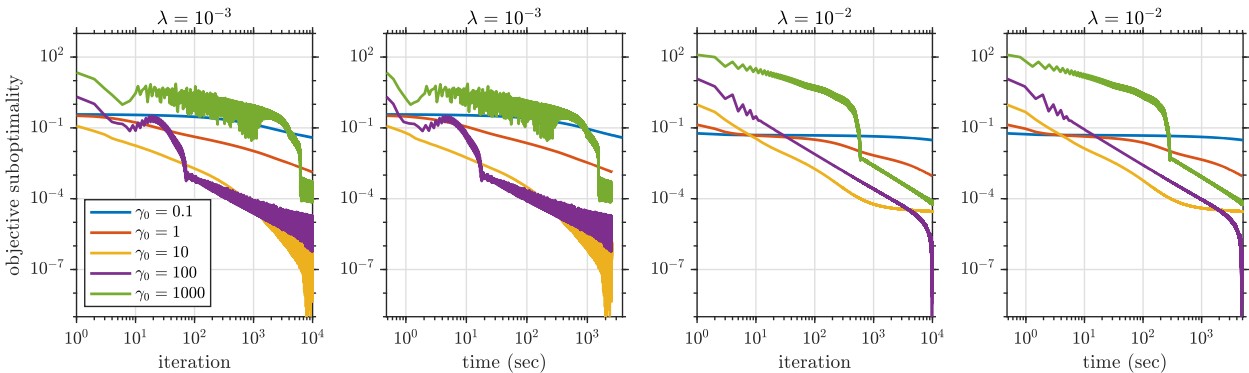

Figure S.4: Empirical performance of ADAPTOS for different choices of $\alpha$ for the overlapping group lasso problem with the hinge-loss. In this experiment we use RCV1 dataset (Lewis et al., 2004) ($n = 677399$, $N = 20242$) and tried two different values of the regularization parameter $\lambda = 10^{-3}$ and $10^{-2}$.

## F.2   Details for Section 6.2

Figure S.6 shows the recovered approximations with $\ell_1$ and $\ell_2$-loss functions along with the original image and the noisy observation. $\ell_1$-loss is known to be more reliable against outliers, and it empirically generates a better approximation of the original image with 26.21 dB peak signal to noise ratio (PSNR) against 21.15 dB for the $\ell_2$-loss.

In Figure S.7 we extend the comparison in Figure 2 with the squared-$\ell_2$ loss.

$$\min_{\boldsymbol{X} \in \mathbb{R}^{m \times n}} \quad \frac{1}{2} \|\mathcal{A}(X) - Y\|_2^2 \quad \text{subject to} \quad \|X\|_* \leq \lambda, \quad 0 \leq X \leq 1, \tag{S.94}$$

Note that the solution set is the same for $\ell_2$ and squared-$\ell_2$ formulations. However, squared-$\ell_2$ loss is smooth whereas $\ell_2$ loss is nonsmooth. Nevertheless, the empirical performance of ADAPTOS for the two formulations are similar. We also compare the evaluation of PSNR over the iterations. This comparison clearly demonstrates the advantage of using the robust $\ell_1$ loss formulation.

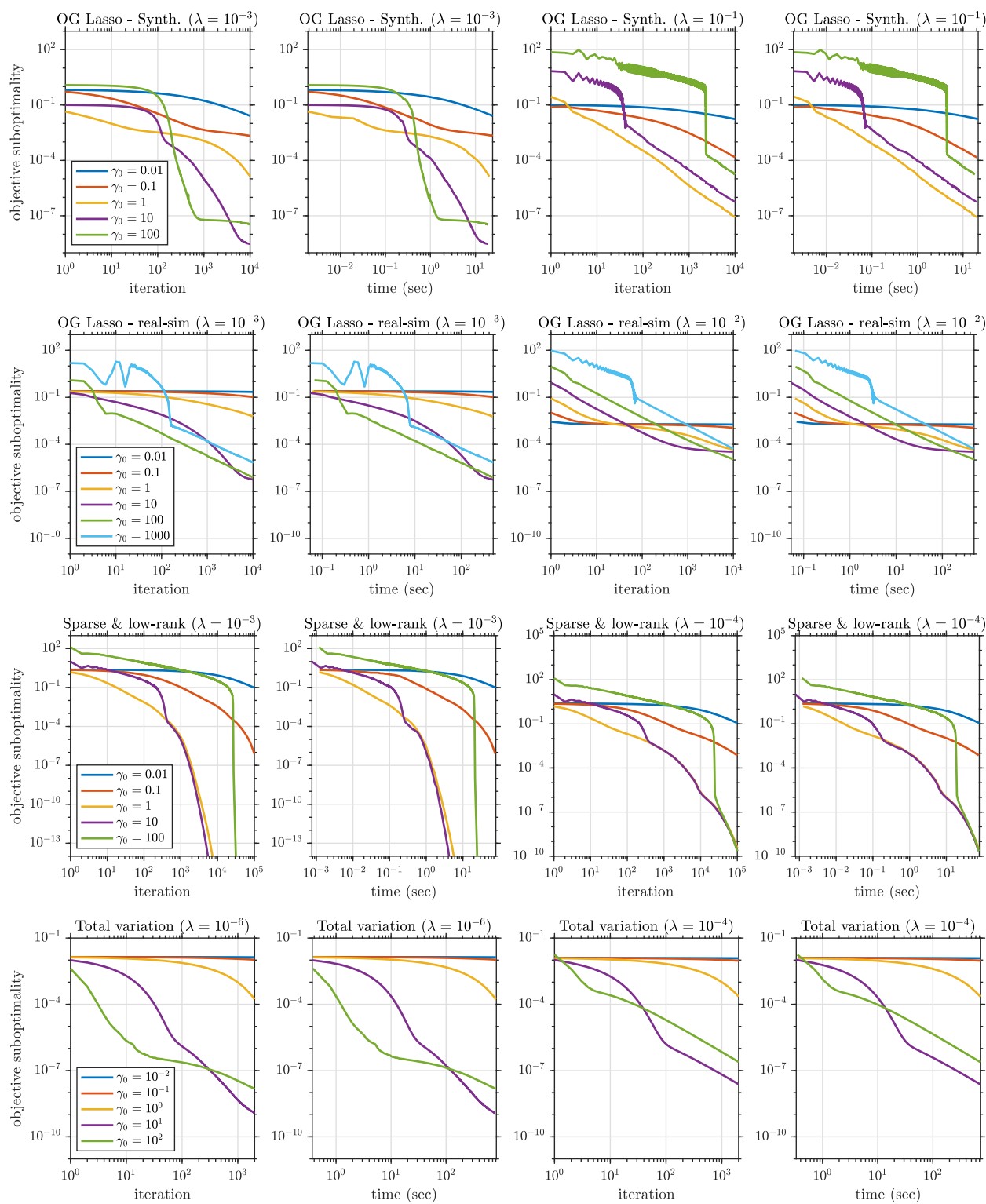

Figure S.5: Empirical performance of ADAPTOS with different choices of $\alpha$ for the problems with smooth and convex loss function studied in Section 6.1.

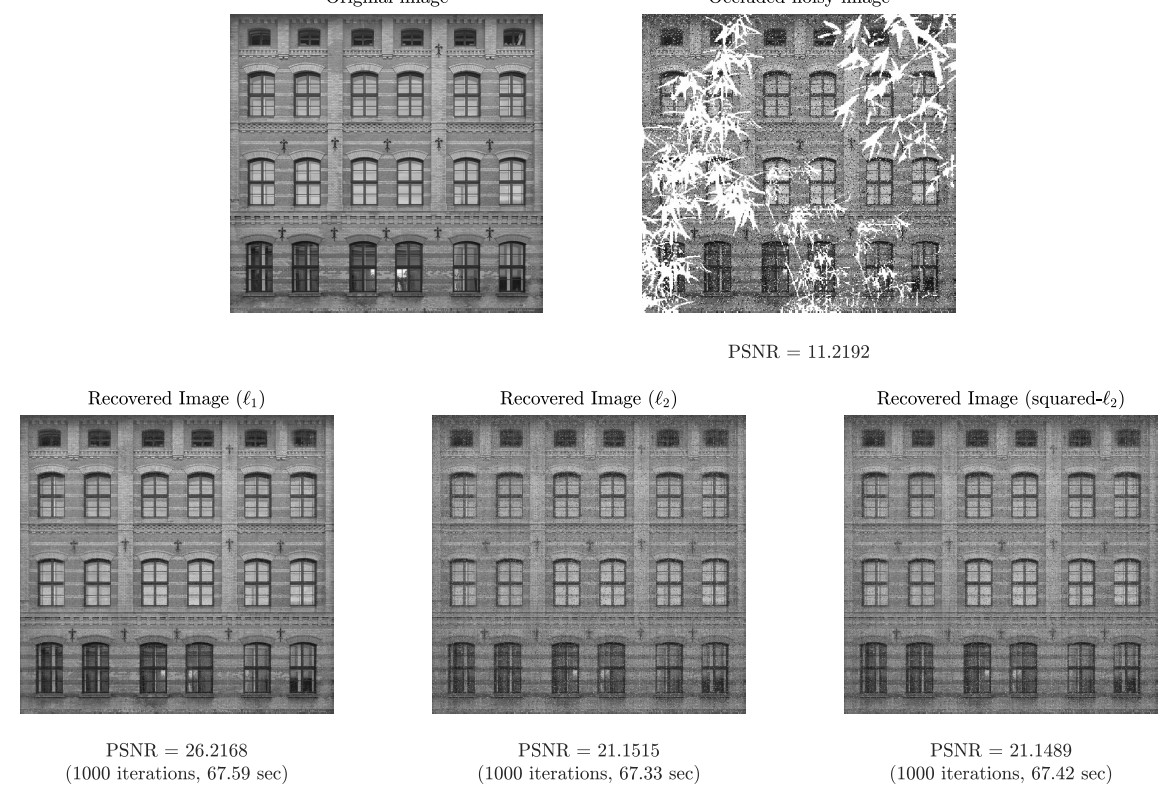

Figure S.6: Comparison or images recovered by minimizing the $\ell_1$, $\ell_2$ and squared-$\ell_2$ loss functions described in Section 6.2. $\ell_1$-loss empirically gives a better approximation with 5dB higher PSNR.

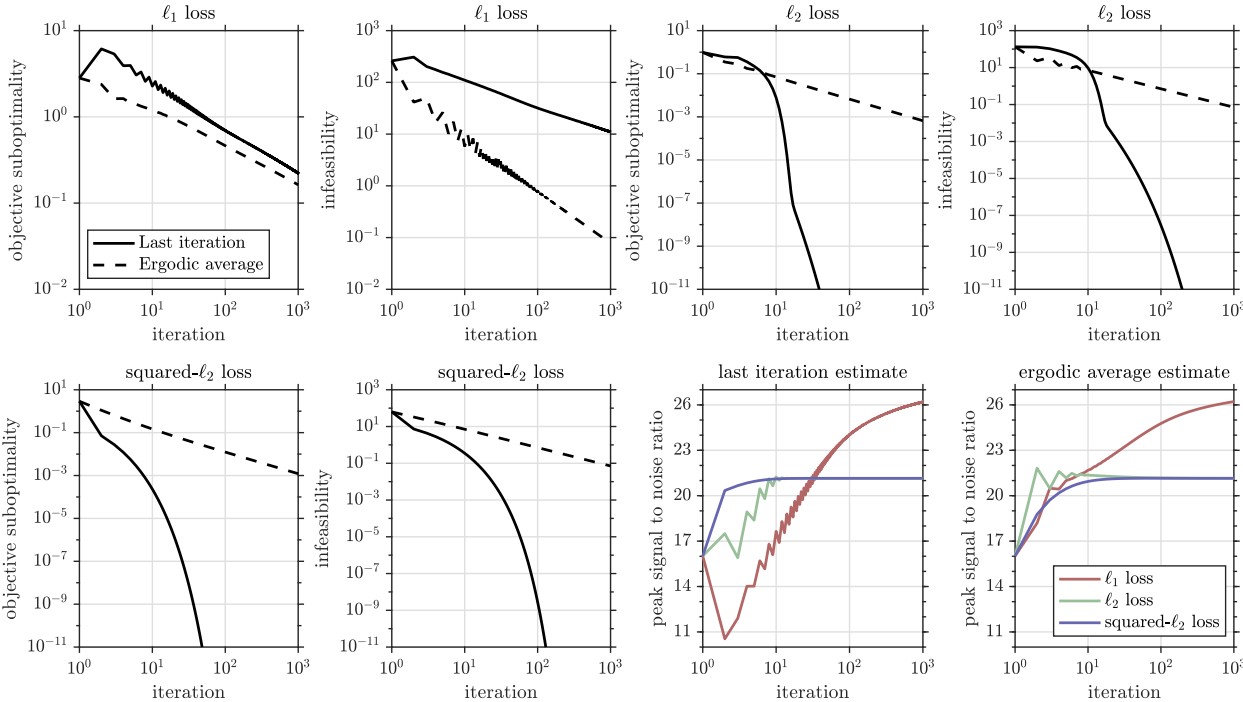

Figure S.7: Performance of ADAPTOS on image impainting problems with $\ell_1$, $\ell_2$, and squared-$\ell_2$ loss functions described in Section 6.2. The performance for nonsmooth $\ell_2$ loss and smooth squared-$\ell_2$ loss are qualitatively similar.

### F.3 Details for Section 6.3

Let $\boldsymbol{y} = f(\boldsymbol{x}, \boldsymbol{w})$ denote a generic deep neural network with $H$ hidden layers, which takes a vector input $x$ and returns a vector output $y$, and $w$ represents the column-vector concatenation of all adaptable parameters. $k$th hidden layer operates on input vector $\boldsymbol{\theta}_k$ and returns $\boldsymbol{\theta}_{k+1}$,

$$\boldsymbol{\theta}_{k+1} = \phi_k(\boldsymbol{W}_k\boldsymbol{\theta}_k + \boldsymbol{b}_k), \quad \text{for } 1 \leq k \leq H, \tag{S.95}$$

where $\boldsymbol{\theta}_1 = \boldsymbol{x}$ denotes the input layer by convention, $\{\boldsymbol{\theta}_k, \boldsymbol{b}_k\}$ are the adaptable parameters of the layer and $\phi_k$ is an activation function to be applied entry-wise. We use ReLu activation (Glorot et al., 2011) for the hidden layers of the network and the softmax activation function for the output layer. We use the same initial weights as in (Scardapane et al., 2017), which is based on the method described in (Glorot & Bengio, 2010).

Given a set of $N$ training examples $\{(\boldsymbol{x}_1, \boldsymbol{y}_1), \ldots, (\boldsymbol{x}_N, \boldsymbol{y}_N)\}$, we train the network by minimizing

$$\min_{\boldsymbol{w} \in \mathbb{R}^n} \quad \frac{1}{N}\sum_{i=1}^{N} L(\boldsymbol{y}_i, f(\boldsymbol{x}_i, \boldsymbol{w})) + \lambda\|\boldsymbol{w}\|_1 + \lambda\sum_{\boldsymbol{\alpha}\in\Omega}\sqrt{|\boldsymbol{\alpha}|}\|\boldsymbol{\alpha}\|_2, \tag{S.96}$$

with the standard cross-entropy loss given by $L(\boldsymbol{y}, f(\boldsymbol{x}, \boldsymbol{w})) = -\sum_{j=1}^{\dim(\boldsymbol{y})} y_j \log(f_j(\boldsymbol{x}, \boldsymbol{w}))$. $\lambda > 0$ is the regularization parameter. We set $\lambda = 10^{-4}$, which is shown to provide the best results in terms of classification accuracy and sparsity in (Scardapane et al., 2017).

The first regularizer ($\ell_1$ penalty) in (S.96) promotes sparsity on the overall network, while the second regularizer (Group-Lasso penalty, introduced in (Yuan & Lin, 2006)) is used to achieve group-level sparsity. The goal is to force all outgoing connections from the same neurons to be simultaneously zero, so that we can safely remove them and obtain a compact network. To this end, $\Omega$ contains the sets of all outgoing connections from each neuron (corresponding to the rows of $\boldsymbol{W}_k$) and single element groups of bias terms (corresponding to the entries of $\boldsymbol{b}_k$).

We compare our methods against SGD, AdaGrad and Adam. We use minibatch size of $400$ for all methods. We use the built-in functions in Lasagne for SGD, AdaGrad and Adam. These methods use the subgradient of the overall objective (S.96). All of these methods have one learning rate parameter for tuning. We tune these parameters by trying the powers of 10. We found that $\gamma_0 = \alpha = 1$ works well for TOS and ADAPTOS. For SGD and AdaGrad, we got the best performance when the learning rate parameter is set to $10^{-2}$, and for Adam we got the best results with $10^{-3}$.

Remark that subgradient methods are known to destroy sparsity at the intermediate iterations. For instance, the subgradients of $\ell_1$ norm are fully dense. In contrast, TOS and ADAPTOS handle the regularizers through their proximal operators. The advantage of using a proximal method instead of subgradients is outstanding. TOS and ADAPTOS result in precisely sparse networks whereas other methods can only get approximately sparse solutions. The comparison becomes especially stark in group sparsity, with no clear discontinuity in the spectrum for other methods.

## G    Additional Numerical Experiments

In this section, we present additional numerical experiments on isotonic regression and portfolio optimization problems. The experiments in this section are performed in MATLAB R2018a with 2.6 GHz Quad-Core Intel Core i7 CPU.

### G.1    Isotonic Regression

In this section, we compare the empirical performance of the adaptive step-size in Section 5 with the analytical step-size in Section 3. We consider the isotonic regression problem with the $\ell_p$-norm loss:

$$\min_{x \in \mathbb{R}^n} \quad \frac{1}{p}\|Ax - b\|_p^p \quad \text{subject to} \quad x_1 \leq x_2 \leq \ldots \leq x_n, \tag{S.97}$$

where $A \in \mathbb{R}^{m \times n}$ is a given linear map and $b \in \mathbb{R}^m$ is the measurement vector. Projection onto the order constraint in (S.97) is challenging, but we can split it into two simpler constraints:

$$\min_{x \in \mathbb{R}^n} \quad \frac{1}{p}\|Ax - b\|_p^p \quad \text{subject to} \quad \begin{Bmatrix} x_1 \leq x_2 \\ x_3 \leq x_4 \\ \vdots \end{Bmatrix} \quad \text{and} \quad \begin{Bmatrix} x_2 \leq x_3 \\ x_4 \leq x_5 \\ \vdots \end{Bmatrix}. \tag{S.98}$$

We demonstrate the numerical performance of the methods for various values of $p \in [1, 2]$. Note that $p = 1$ and $p = 2$ capture the nonsmooth least absolute deviations loss and the smooth least squares loss respectively. For larger values of $p$, we expect ADAPTOS to exhibit faster rates by adapting to the underlying smoothness of the objective function.

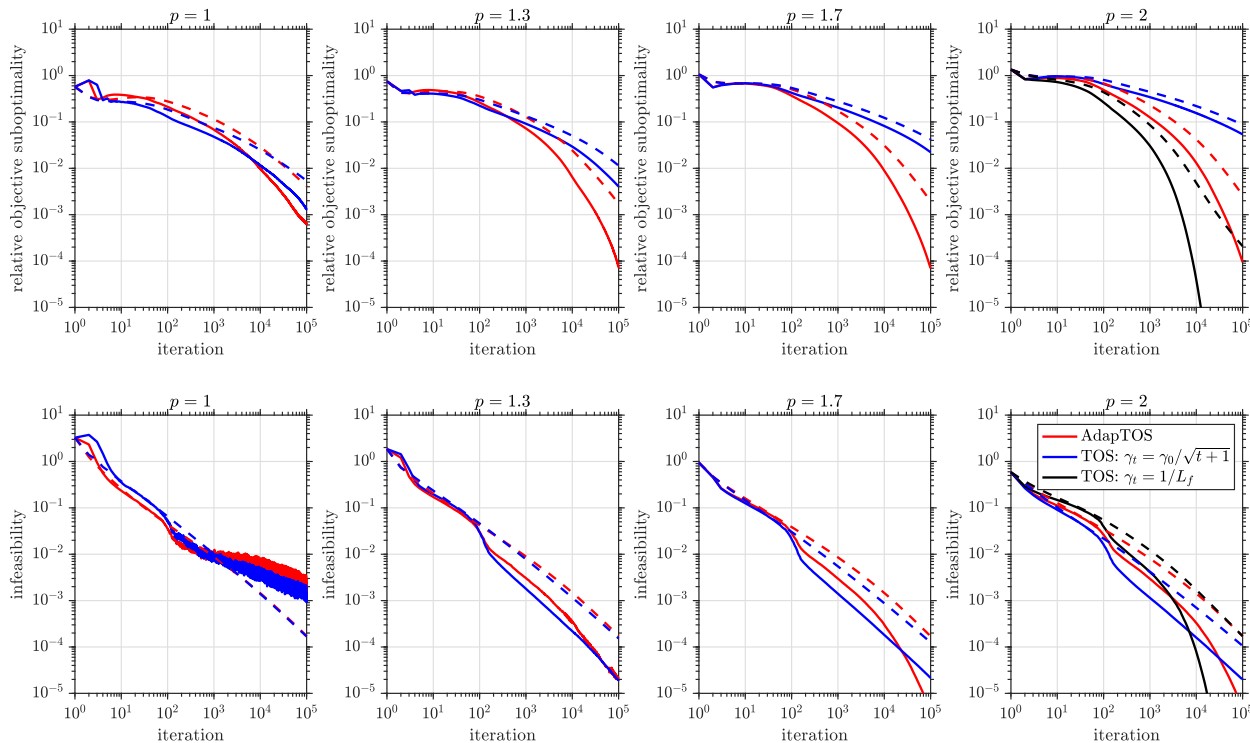

Figure S.8: Comparison of the empirical performance of TOS with the analytical step-size in Section 3 and the adaptive step-size in Section 5 on the isotonic regression problem in (S.97) with the $\ell_p$-loss function for various $p \in [1, 2]$. For larger values of $p$, ADAPTOS exhibits faster convergence rates by adapting to the underlying smoothness of the objective function. For $p = 2$, the problem is smooth so we also consider the standard fixed step-size $\gamma = 1/L_f$ in this setting. Solid lines represent the last iteration and the dashed lines correspond to the ergodic sequences $\bar{x}_t$ and $\bar{z}_t$.

We generate a synthetic test setup. To this end, we set the problem size as $m = 100$ and $n = 200$. We generate right and left singular vectors of $A$ by computing the singular value decomposition of a random matrix with *iid* entries drawn from the standard normal distribution. Then, we set the singular values according to a polynomial decay rule such that the $i$th singular vector is $1/i$. We generate $x^\natural \in \mathbb{R}^n$ by sorting $n$ *iid* samples from the standard normal distribution. Then, we compute the noisy measurements $b = Ax^\natural + 0.1\xi$ where the entries of $\xi$ is drawn *iid* from the standard normal distribution.

By considering a decaying singular value spectrum for $A$ we control the condition number and make sure the problem is not very easy to solve. By adding noise, we ensure that the solution is not in the relative interior of the feasible set. Therefore, this experiment also supports our claim that ADAPTOS can achieve fast rates when the objective is smooth even if the solution does not lie in the interior of the feasible set.

When the problem is nonsmooth, *i.e.*, when $p < 2$, we use TOS with the analytical step-size in Section 3 and the adaptive step-size in Section 5. We choose $\alpha = \beta = \gamma_0 = 1$ without any tuning. When $p = 2$, the problem is smooth so we also try TOS with the standard constant step-size $\gamma = 1/L_f$ in this setting. We run each algorithm for $10^5$ iterations. In order to find the ground truth $f_\star$ we solve the problem to very high precision by using CVX (Grant & Boyd, 2014) with the SDPT3 solver (Toh et al., 1999).

We repeat the experiments with 20 randomly generated data with different seeds and report the average performance in Figure S.8. This figure compares the performance we get by different step-size strategies in terms of objective suboptimality ($|f(z_t) - f_\star|/f_\star$) and infeasbility bound ($\|z_t - x_t\|$). As expected, ADAPTOS performs better as $p$ becomes larger. Although it does not exactly match the performance of the fixed step-size $1/L_f$ when $f$ is smooth ($p = 2$), remark that ADAPTOS does not require any prior knowledge on $L_f$ or $G_f$.

## G.2 Portfolio Optimization

In this section, we demonstrate the advantage of stochastic methods for machine learning problems. We consider the portfolio optimization with empirical risk minimization from Section 5.1 in (Yurtsever et al., 2016):

$$\min_{x \in \mathbb{R}^n} \quad \frac{1}{2} \sum_{i=1}^{N} |\langle a_i, x \rangle - b|^2 \quad \text{subject to} \quad x \in \Delta \quad \text{and} \quad \langle a_{\text{av}}, x \rangle \geq b \tag{S.99}$$

where $\Delta$ is the unit simplex. Here $n$ is the number of different assets and $x \in \Delta$ represents a portfolio. The collection of $\{a_i\}_{i=1}^{N}$ represents the returns of each asset at different time instances, and the $a_{\text{av}}$ is the average returns for each asset that is assumed to be known or estimated. Given a minimum target return $b \in \mathbb{R}$, the goal is to reduce the risk by minimizing the variance. As in (Yurtsever et al., 2016), we set the target return as the average return over all assets, *i.e.*, $b = \text{mean}(a_{\text{av}})$.

In addition, we also consider a modification of (S.99) with the least absolute deviation loss, which is nonsmooth but known to be more robust against outliers:

$$\min_{x \in \mathbb{R}^n} \quad \sum_{i=1}^{N} |\langle a_i, x \rangle - b| \quad \text{subject to} \quad x \in \Delta \quad \text{and} \quad \langle a_{\text{av}}, x \rangle \geq b. \tag{S.100}$$

We use 4 different real portfolio datasets: Dow Jones industrial average (DJIA, 30 stocks for 507 days), New York stock exchange (NYSE, 36 stocks for 5651 days), Standard & Poor's 500 (SP500, 25 stocks for 1276 days), and Toronto stock exchange (TSE, 88 stocks for 1258 days).[2]

For both problems and each dataset, we run ADAPTOS with full (sub)gradients and stochastic (sub)gradients and compare their performances. We choose $\alpha = \beta = 1$ without tuning and run the algorithms for 10 epochs. In the stochastic setting, we evaluate a (sub)gradient estimator from a single datapoint chosen uniformly at random with replacement at every iteration. We run the stochastic algorithm 20 times with different random seeds and present the average performance. To find the ground truth $f_\star$ we solve the problems to very high precision by using CVX (Grant & Boyd, 2014) with the SDPT3 solver (Toh et al., 1999). Figures S.9 and S.10 present the results of this experiment for (S.99) and (S.100) respectively.

---

[2] These four datasets can be downloaded from http://www.cs.technion.ac.il/~rani/portfolios/

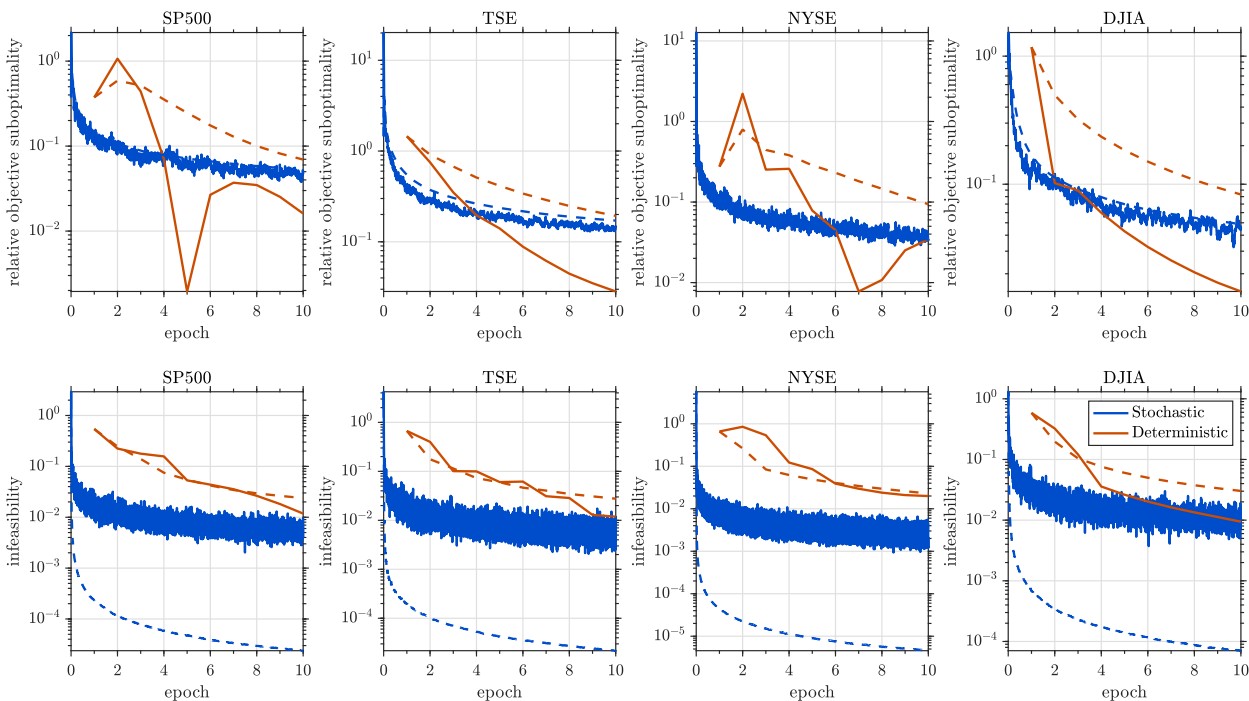

Figure S.9: Comparison of ADAPTOS with stochastic and deterministic gradients on the smooth portfolio optimization problem with least-squares loss in (S.99) for four different datasets. Solid and dashed lines represent the last and ergodic iterates respectively.

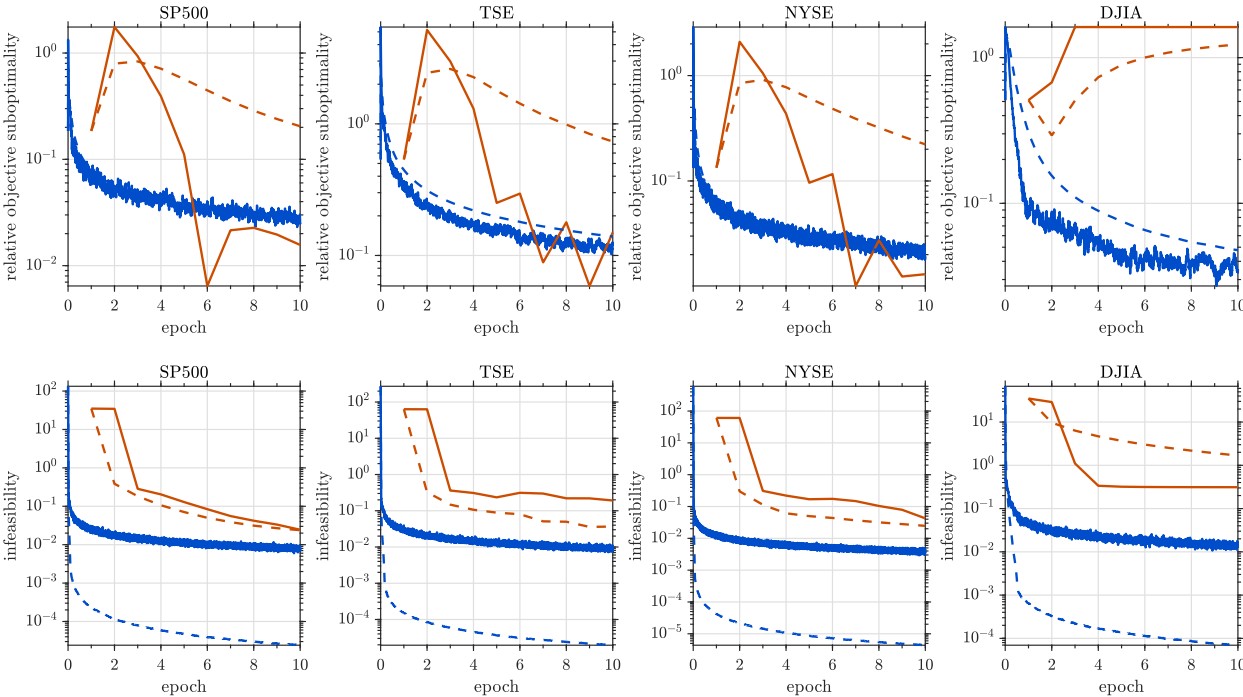

Figure S.10: Comparison ADAPTOS with stochastic and deterministic gradients on the nonsmooth portfolio optimization problem with least absolute deviations loss in (S.100) for four different datasets. Solid and dashed lines represent the last and ergodic iterates respectively.