# OpenReview forum: "Three Operator Splitting with Subgradients, Stochastic Gradients, and Adaptive Learning Rates"
_NeurIPS.cc/2021/Conference — NeurIPS 2021 Poster_

### Official Review · Reviewer_7qvs · 2021-07-01

**Rating:** 6
**Confidence:** 4

**Summary:**

In this paper, the authors analyze a variant of the classical three operator splitting algorithm where the smooth gradient (or co-coercive operator) is replaced by a subgradient (which is only a  monotone operator). This enables the minimization of the sum of three functions, two of which have explicit proximity operators and one is accessed through a subgradient.

**Main Review:**

First of all, I would say that the main reason this case has not been covered previously is that with a subgradient, the TOS operator (ie (2) with u as in Lemma 1) is no longer firmly non-expansive which means that most of the usual operator theory for (Fejér monotonous) iterates convergence breaks down. If my intuition is correct maybe a sentence about that would be interesting for the reader.

* I understand the need to have a generic u_t in algorithm 1 but could you give the reader the intuition that it should be a gradient of f (in original TOS) or something related (maybe as a comment).

* In the proof of Lemma 1, you need Assumption 1 (for S.3 to hold) but it is not explicitly put in the lemma or the other theorems. Should this be a blanket assumption? (eg. in Theorem 1, you will need Lemma 1 but only assumption is put explicitly)


* I am a bit troubled by the formulation of Lemma 1 since TOS is not R^n to R^n but takes two values. Replacing u directly by its (set-)value and consider the resulting set-valued operator would be in line with standard monotone operator theory. (and the fact that it may not be single valued would show the difference with the usual case).

* I find Theorem 1 slightly deceiving as x_T is present in h while z_T is present in the other, I would be nice to have a warning. It would also be nice to have a sketch of the proof. Basically, I think that Supp. C builds on operator theory to get boundedness which is the only possible out come without smothness of the gradient and then in D, you explicitly use the functions to get a functional convergence.

* I also missed at first read that the adaptive part only focused on indicator functions, maybe the first sentence of Sec 5 should be emphasized. Also, what are the technical limitations that prevent from considering proximity operators there?

* In the numerical experiments, I am a bit dubious about the the smooth case of Figure 1. You use AdapTOS for smooth problems (except the sparse low-rank) which is outside of the theoretical framework right? Also, the LS version seem to perform worse than the vanilla counterparts in some experiments is that normal? In the nonsmooth Sparse Low Rank case, I would have expected to see also the policy of Theorem 1 for comparison.

Minor comments:
* line 16:  R \cap \{\infty\} should be  R \cup \{\infty\}
* Caption of Fig. 1  "Lipchitz constant the regularizer" ?
* in S.29, you have \gamma but above, line 80, you have gamma_t

Post-rebuttal: Despite some limitations and clarifications to be made, the paper has merits.

**Time Spent Reviewing:**

3

---

> ### Author Response · Authors · 2021-08-10
> **Thank you for your feedback.**
>
> **Technical challenge and the proof sketch**
>
> Thank you for carefully reading our paper. Your intuition is correct: We lose Fejer monotonicity with the subgradient step and the usual approach breaks down. Instead, we show that the iterates of TOS remain bounded, which is weaker than Fejer monotonicity but still holds for the subgradient extension. The boundedness of the iterates plays a key role in the analysis. We will add a proof sketch in the revision.
>
> **Other comments**
> - We state in lines 110-111, right before (2), that $u_t = \nabla f (z_t)$ yields the original TOS algorithm. We will emphasize this more (and earlier in the text, when Algorithm 1 is introduced) in the revision.
> - Assumption 1 is a blanket assumption, but we agree that this is not very clear in the draft. We will make this explicit in the revision.
> - Indeed, we can formulate Lemma 1 with set-valued operators; as $y \in \texttt{TOS}(y, \partial f(\mathrm{prox}_{\gamma g}(y))$ iff $y$ is a solution.
> - We will add a warning between Theorem 1 and Corollary 1 to emphasize that $f+g$ and $h$ are evaluated at different points. This motivates Corollary 1.
> - We indicate that AdapTOS is for optimizing a convex loss over the intersection of convex sets, an important subclass of Problem (1), in lines 10-12 in the abstract, lines 45-49 in the introduction, lines 174-175 in Section 5, and lines 303-304 in the Conclusions. We will clarify these statements.
> In the regularized setting, we need $\gamma_t$ in the first step (to find $z_t$) before we have access to the (sub)gradient ($u_t$) (note that $\gamma_t$ depends on $||u_t||$.) This is the main challenge that prevents us from extending these results (see line 183).
> - AdapTOS can be used for smooth problems. It guarantees $\tilde{\mathcal{O}}(1/\sqrt{t})$ convergence rate if the gradient is bounded (which holds automatically for smooth $f$ if the domain is bounded). Or, if the solution remains in the interior of the domain, then AdapTOS guarantees $\tilde{\mathcal{O}}(1/t)$ rate. However, AdapTOS in Figure 1 is used outside of its theoretical framework because $g$ and $h$ in these examples are not indicator functions.
> - We observe that TOS with line-search (TOS-LS) performs better when the regularization parameter is small, but the standard TOS can outperform TOS-LS when the regularization parameter is large. This is in accordance with the tutorial examples in COPT Python Library. See, for example, the “Group Lasso with Overlap” tutorial on the webpage of the COPT package (homepage > Example Gallery > Group lasso with overlap).
> - We can add more experiments to compare AdapTOS with the step-size policy of Theorem 1. Note that Huber loss is also smooth.

---

### Official Review · Reviewer_eqKr · 2021-07-16

**Rating:** 6
**Confidence:** 5

**Summary:**

The paper considers three extension of the three operator splitting (TOS) algorithm: sub-gradient instead of gradient, stochastic  gradient and adaptive step-sizes. Convergence rates are established for each extension.


**Limitations And Societal Impact:**

See above for limitations and no negative societal impact.


**Main Review:**

First of all, the paper missed a recent work “Three Operator Splitting with a Nonconvex Loss Function, A. Yurtsever et al 2021’’ which studied TOS in the non-convex setting and with stochastic gradient. For example, problem of type (15) was studied in this work.

Due to the above missing reference, the main strength of the paper is TOS with sub-gradient. However the proposed algorithm is not what I expected when reading lines 34-36 of the introduction. It appears to me that the authors would replace the computation of one prox-operator with sub-gradient descent. However, the proposed algorithm still needs computing two prox-operators, and gradient descent part is relaxed.

The convergence results of the paper match existing result of sub-gradient descent and stochastic gradient.

The proposed AdapTOS is difficult to understand, as the step-size is motivated by adaptive stochastic gradient method, and there is no randomness here.  For problem (27) considered in the numerical experiments, it would be great if the authors comparing the proposed with other methods where no sub-gradient descent, such as ADMM type methods or Primal-Dual splitting.


**Time Spent Reviewing:**

4

---

> ### Author Response · Authors · 2021-08-10
> **Thank you for your feedback.**
>
> **Comparison against (Yurtsever et al., 2021)**
>
> Thank you for this reference. We will present a detailed comparison against this paper in the revision. However, there are important distinctions between this paper and our work: \
> 1- First of all, this paper focuses on non-convex $f$ but *requires a stringent assumption that the domain is bounded*. We assume $f$ is convex but we do not have the boundedness assumption. Hence, they do not fully cover our problem template. \
> 2- They prove $\mathcal{O}(1/t^{1/3})$ convergence rate with an increasing mini-batch size (i.e., they require the variance to be reduced over the iterations) which leads to $\mathcal{O}(1/\epsilon^5)$ stochastic first-order oracle complexity for finding an $\epsilon$-suboptimal solution. In contrast, we prove $\mathcal{O}(1/t^{1/2})$ convergence rate with a fixed mini-batch size, which yields $\mathcal{O}(1/\epsilon^2)$ stochastic first-order oracle complexity.
>
> To our knowledge, our paper presents the first analysis for stochastic TOS without strong convexity or variance reduction. *Given the orders of magnitude improvement in the stochastic gradient complexity, we kindly ask you to reconsider your evaluation of the significance of our paper in the stochastic setting.*
>
> **On the non-smooth extension**
>
> Please note that replacing one of the prox-updates with a subgradient step is not interesting because it yields some form of the proximal subgradient method. Given an optimization problem, the practitioner can select how to assign the individual terms $f$, $g$, and $h$. However, the original TOS algorithm only allows non-smooth terms to be accessed via their proximal operators (hence, by assigning them as $g$ or $h$). In contrast, our method allows handling a non-smooth term via subgradients (by assigning it to $f$), which is considerably more flexible. Note that the sum of (sub)gradients is a (sub)gradient of the sum. Hence only one (sub)gradient step is enough. Similarly, two proximal steps are enough to formulate any finite number of non-smooth terms via their individual prox-operators by using the product space technique (see Section 6.1 in (Briceño-Arias, 2015)). Therefore, our method can solve problems of the form $\min_x \sum_{i=1}^m \psi_i (x)$ for any finite m > 0 by using the individual (sub)gradients or the proximal operators of each $\psi_i$ term. We will clarify this in the revision.
>
> L. M. Briceño-Arias. Forward-Douglas–Rachford splitting and forward-partial inverse method for solving monotone inclusions. Optimization, 64(5):1239–1261, 2015.
>
> **Intuition for the AdapTOS step-size**
>
> The step-size for AdapTOS is motivated by the prior work on adaptive and universal gradient methods (see lines 44-45 and 179-180). For the closest example, see (Levy, 2017): They adopt a similar step-size for designing adaptive (sub)gradient methods. These methods attain universal convergence rates, that is, the same step-size guarantees $\tilde{\mathcal{O}}(1/t)$ convergence rate when the objective function is smooth and $\tilde{\mathcal{O}}(1/\sqrt{t})$ when the objective is non-smooth or the gradient is stochastic. Note that we also consider the stochastic setting in this section, see Theorem 5.
>
> Here is the rough idea: When the problem is smooth and deterministic, the gradient norm quickly diminishes near a solution hence the step-size is almost constant. On the other hand, if the problem is non-smooth (or stochastic), then the subgradient (resp. stochastic gradient) norm may not diminish near the solution, leading to the use of decreasing step-size.
>
> **Other comments**
>
> We can add comparisons with ADMM-type methods and primal-dual operator splitting methods in the revision.

---

> > ### Comment · Reviewer_eqKr · 2021-08-24
> > **Response to Rebuttal**
> >
> > Sounds fair to me, and please clarify these in the revision.

---

### Official Review · Reviewer_fPkS · 2021-07-18

**Rating:** 7
**Confidence:** 4

**Summary:**

This paper provides an extension of the Three Operator Splitting methods for composite optimization, of the form $f + g + h$. Previous work assumed $f$ to be smooth. This work tackles two cases: 1) the case where $f$ is non-smooth, and the case where additionally, only a stochastic estimator of the (sub)gradient of $f$ is available. It can be seen as an extension of (Yurtsever et al 2018) in the non-smooth, non-strongly convex case.
The paper provides a theoretical worst-case analysis of convergence, and gives non-asymptotic rates for these two cases. It also proposes an analyses an adaptive variant, where step size depends on the previous observed subgradient magnitudes, a la AdaGrad. It provides experimental validation of the method on a non-smooth problem, and on a non-convex non-smooth problem.

**Main Review:**

The paper is clear and well written. Extensions of the TOS method in the non-smooth setting should be interesting to the community.

Additional experiments in the 1st setting (robust recovery) in the stochastic setting (large images + subsampling pixels to compute the [sub]gradients) would be interesting to evaluate scalability of the stochastic method vs non-stochastic.

Question on the Image recovery experiment: I see in the code that the loss is not the squared loss for the L2 case. Why is that? Since the squared L2 is strongly convex, and the two problems are equivalent, the squared loss would always be used in practice. Also, it allows to avoid numerical instability due to dividing by the norm of $\mathcal A(x) - y$.

Question on ResNet experiments: what sparsity structure did you try to enforce in your preliminary experiments ? Using individual convolutional filters as groups for the group LASSO (and not penalizing/regularizing the fully connected layer) may be beneficial in this context. Also, keeping some type of L2 regularization (weight decay) should also be helpful. I would be greatly interested in these results.

The latest commit on Lasagne was in 2015... Perhaps consider writing the AdapTOS optimizer in PyTorch, Jax or Tensorflow to promote its usage, and reproducibility?

Comments on the form:
Typo: line 115 unnecessary "a"
Typo: line 202: we believe this to be a limitation...

**Time Spent Reviewing:**

2

---

> ### Author Response · Authors · 2021-08-10
> **Thank you for your feedback.**
>
> **Experiments on robust recovery in the stochastic setting**
>
> We can add more experiments on robust recovery to evaluate the scalability of the stochastic method vs non-stochastic.
>
> **Squared $\ell_2$ loss**
>
> Indeed, we tried both squared and non-squared $\ell_2$ loss for this problem. The results are similar. We chose to present the non-squared $\ell_2$ loss to demonstrate that TOS can exhibit fast (locally linear) convergence even for some non-smooth $f$. We can add the results for the squared $\ell_2$ loss in the revision for comparison.
>
> **ResNet experiments**
>
> We tried using individual convolutional filters as groups but we also regularized the fully connected layer. We did not consider an $\ell_2$ regularization. We can include the code for these preliminary experiments in the supplements for interested readers.
>
> **Other comments**
>
> We will consider writing our experiments in Section 6.3 for a modern framework like Jax, TensorFlow, or PyTorch.

---

> > ### Comment · Reviewer_fPkS · 2021-08-30
> > **Response to rebuttal**
> >
> > Looks good to me, thank you. My score is unchanged.

---

### Official Review · Reviewer_ofGn · 2021-07-19

**Rating:** 7
**Confidence:** 3

**Summary:**

This paper proposes to extend the three operator splitting (TOS) algorithm via stochastic gradients, subgradients and adaptive step-sizes respectively. Convergence rates and numerical experiments of these extensions are provided.

**Limitations And Societal Impact:**

While the author have discussed the limitations of their work, I do expect numerical experiments to illustrate the theoretical results in Section 5.

**Main Review:**

Originality:

The application of subgradients in a TOS scheme seems to be original to me. The use of stochastic gradients in a TOS scheme is well mentioned in Section 2 of the paper.


Quality and significance:

The quality of this paper is above average. I also believe that the proposed extensions are significant to study.


Clarity:

I think some statements in the paper require clarifications as they are misleading. For instance, in the abstract, point (ii) has led me to think that the authors are going to replace the functions which we evaluate their proximity operators on (i.e., $g$ and $h$) by their subgradients. But after looking into the paper, the authors are replacing $\nabla f$ by $\partial f$ instead, where the smooth function $f$ in the original TOS becomes nonsmooth. The subsection titles of numerical experiments are also misleading: these problems are also nonsmooth because of the nonsmooth regularizers.

Typos:
* Line 16: $\mathbb{R}\cup\lbrace +\infty\rbrace$ instead of $\mathbb{R}\cap\lbrace +\infty\rbrace$
* Line 120: *splitting* instead of slitting
* Line 143: redundant *an*
* Line 233: $X$ instead of X
* Line 284: 93% instead of %93

**Post-rebuttal:**
I have read the author response and decided to keep my evaluation unchanged, given that the authors agree to clarify the misleading notions in the original manuscript.

**Time Spent Reviewing:**

4

---

> ### Author Response · Authors · 2021-08-10
> **Thank you for your feedback.**
>
> **On the non-smooth extension**
>
> Please note that replacing one of the prox-updates with a subgradient step is not interesting because it yields some form of the proximal subgradient method. Given an optimization problem, the practitioner can select how to assign the individual terms $f$, $g$, and $h$. However, the original TOS algorithm only allows non-smooth terms to be accessed via their proximal operators (hence, by assigning them as $g$ or $h$). In contrast, our method allows handling a non-smooth term via subgradients (by assigning it to $f$), which is considerably more flexible. Note that the sum of (sub)gradients is a (sub)gradient of the sum. Hence only one (sub)gradient step is enough. Similarly, two proximal steps are enough to formulate any finite number of non-smooth terms via their individual prox-operators by using the product space technique (see Section 6.1 in (Briceño-Arias, 2015)). Therefore, our method can solve problems of the form $\min_x \sum_{i=1}^m \psi_i (x)$ for any finite m > 0 by using the individual (sub)gradients or the proximal operators of each $\psi_i$ term. We will clarify this in the revision.
>
> L. M. Briceño-Arias. Forward-Douglas–Rachford splitting and forward-partial inverse method for solving monotone inclusions. Optimization, 64(5):1239–1261, 2015.
>
> **Other comments**
>
> We will correct the subsection titles, and we can add more numerical experiments in the revision to illustrate the theoretical results in Section 5.

---

> > ### Comment · Reviewer_ofGn · 2021-08-19
> > **Response to Rebuttal**
> >
> > I am not suggesting "replacing one of the prox-updates with a subgradient step". It is just that the use of misleading notions would confuse readers, so clarifications and/or a more accurate use of notions are needed.

---

### Author Response · Authors · 2021-08-10
**Author response**

We would like to thank all reviewers for their efforts and considered thoughts in reviewing our paper. We are grateful for your feedback and suggestions to improve our paper. We present our detailed discussions in separate messages to each reviewer.

---

### Decision · Program_Chairs · 2021-09-27

**Decision:**

Accept (Poster)

**Comment:**

All reviewers agree that this paper is worthy of publication. Although some reviewers had initial remarks regarding missing reference, the authors have provided a strong rebuttal that addressed these concerns.

I encourage the authors to take into account the feedback provided by the reviewers, in particular:

  * Reviewers 7qvs, ofGn and eqKr make excellent clarification suggestions on motivation and presentation.

   * Reviewer fPkS suggests some improvements to the experimental section.

   * Reviewer eqKr raises the issue of an important missing reference. This review also raises issues regarding clarity, motivation and comparison with other methods.